# Repeated intravenous administration of hiPSC-MSCs enhance the efficacy of cell-based therapy in tissue regeneration

Si-Jia Sun [1,2,9], Fei Li[2,9], Ming Dong[3], Wei-Hao Liang [4], Wing-Hon Lai[2], Wai-In Ho [2], Rui Wei[2], Yan Huang[5], Song-Yan Liao [2,6✉] & Hung-Fat Tse[2,6,7,8✉]

We seek to demonstrate whether therapeutic efficacy can be improved by combination of repeated intravenous administration and local transplantation of human induced pluripotential stem cell derived MSCs (hiPSC-MSCs). In this study, mice model of hind-limb ischemia is established by ligation of left femoral artery. hiPSC-MSCs ($5 \times 10^5$) is intravenously administrated immediately after induction of hind limb ischemia with or without following intravenous administration of hiPSC-MSCs every week or every 3 days. Intramuscular transplantation of hiPSC-MSCs ($3 \times 10^6$) is performed one week after induction of hind-limb ischemia. We compare the therapeutic efficacy and cell survival of intramuscular transplantation of hiPSC-MSCs with or without a single or repeated intravenous administration of hiPSC-MSCs. Repeated intravenous administration of hiPSC-MSCs can increase splenic regulatory T cells (Tregs) activation, decrease splenic natural killer (NK) cells expression, promote the polarization of M2 macrophages in the ischemic area and improved blood perfusion in the ischemic limbs. The improved therapeutic efficacy of MSC-based therapy is due to both increased engraftment of intramuscular transplanted hiPSC-MSCs and intravenous infused hiPSC-MSCs. In conclusion, our study support a combination of repeated systemic infusion and local transplantation of hiPSC-MSCs for cardiovascular disease.

[1] Division of Cardiology, Department of Medicine, The First Affiliated Hospital of Soochow University, Soochow University, Suzhou, China. [2] Cardiology Division, Department of Medicine, Queen Mary Hospital, the University of Hong Kong, Hong Kong SAR, China. [3] Bioland Laboratory, Guangzhou Regenerative Medicine and Health Guangdong Laboratory, Guangzhou, China. [4] Department of Cardiology, The First Affiliated Hospital of Sun Yat-Sen University, Guangzhou, China. [5] Anhui Provincial laboratory of inflammatory and immunity disease, Institute of Innovative Drugs, School of Pharmacy, Anhui Medical University, Hefei, China. [6] Shenzhen Institutes of Research and Innovation, the University of Hong Kong, Hong Kong SAR, China. [7] Hong Kong-Guangdong Joint Laboratory on Stem Cell and Regenerative Medicine, the University of Hong Kong and Guangzhou Institutes of Biomedicine and Health, Hong Kong SAR, China. [8] Department of Medicine, Shenzhen Hong Kong University Hospital, Shenzhen, China. [9] These authors contributed equally: Si-Jia Sun, Fei Li. ✉email: lsy923@hku.hk; hftse@hku.hk

Regenerative therapies, characterized by transplantation of pluripotent stem cells or committed cells to repair or replace damaged tissues, have been widely utilized in the treatment of cardiovascular diseases such as myocardial infarction (MI) and peripheral artery diseases[1–3]. Recent evidence showed that cells harvested from elderly sick patients elicited compromised proliferation and angiogenesis potential that could evoke a compromised therapeutic effect on tissue regeneration and repair[4]. Therefore, allogenic pluripotent stem cell derived cell types with high therapeutic potential and an "off-the-shelf" cell sources hold greater appeal to clinical investigators, rather than cells from those patients[5,6]. Our previous studies demonstrated that intramuscular transplantation of human induced pluripotent stem cell (hiPSC) derived mesenchymal stromal cells (hiPSC-MSCs) improved blood perfusion in a mouse model of hind-limb ischemia through promoting angiogenesis and reducing macrophage infiltration[7]. However, the retention of transplanted cells was remarkably low albeit standard immunosuppression therapies[7,8]. The limited engraftment of local transplanted cells hurdles therapeutic application of cell-based therapies.

The immunomodulatory effects of MSCs have been widely discussed for a generation. Recently, a wide spectrum of studies showed that pre-transplantation systemic administration of MSCs prolonged the survival of allograft in solid organ transplantation[9–11]. Our recent study also showed that pre-transplantation systemic intravenous administration of hiPSC-MSCs improved the survival and the therapeutic efficacy of intramyocardial transplanted hiPSC-MSCs and hiPSC derived cardiomyocytes (hiPSC-CMs) in a mouse MI model[12]. The improved cellular engraftment was associated with the immunomodulatory effects of intravenous hiPSC-MSCs preconditioning[12]. Moreover, previous study reported that systemic infused MSCs contributed to improvement of cardiac function and repeated dosing have a superior therapeutic effects than a single administration[13–16]. Therefore, it is of great interest to investigate whether superior immunomodulatory effects of MSC-based therapy can be achieved by combination of repeated cellular infusion and local cellular transplantation.

In this study, we hypothesized that a combination of repeated systemic intravenous hiPSC-MSCs infusion and intramuscular hiPSC-MSCs transplantation would elicit an enhanced immunomodulatory response in a mouse model of hind-limb ischemia, compared with intramuscular hiPSC-MSCs transplantation with or without single intravenous hiPSC-MSC infusion. The potential immunomodulatory ability of repeated intravenous hiPSC-MSCs infusion every week or every 3 days to enhance the survival and engraftment of intramuscular transplanted hiPSC-MSCs was also investigated.

## Results

### The therapeutic efficacy of intravenous hiPSC-MSCs infusion without intramuscular cellular transplantation.
First, we determined whether hiPSC-MSCs could migrate into the ischemic limb after a single intravenous cellular infusion. Our results showed that most of the hiPSC-MSCs engrafted into the liver 12 h after infusion (Supplementary Fig. 1). The engrafted hiPSC-MSCs gradually migrated into the ischemic limb at day 3 and disappeared at day 14 (Supplementary Fig. 1). A few cells engrafted in the ischemic limb, the engraftment rate was extremely low, evidenced by the DiR signal that was $9.8 \times 10^6$ at day 7 after a single intravenous administration of $5 \times 10^5$ hiPSC-MSCs versus $1.4 \times 10^9$ 7 days after a single intramuscular injection.

To compare intravenous cellular administration and intramuscular cellular delivery, three groups of mice that received intravenous hiPSC-MSC infusion once, every week or every 3 days without intramuscular administration of hiPSC-MSCs respectively and one group that received intramuscular hiPSC-MSC delivery only were employed (Fig. 1a). Intravenous administration of hiPSC-MSCs once, every week or every 3 days without intramuscular administration of hiPSC-MSCs in the Saline-MSC/once, Saline-MSC/week and Saline-MSC/3 days groups significantly improved blood perfusion from day 7 onwards compared with the ischemia group (Fig. 1b, all $p < 0.05$). Repeated intravenous administration of hiPSC-MSCs in the Saline-MSC/week and Saline-MSC/3 days groups further increased blood perfusion at day 35 compared with the Saline-MSC/once group (Fig. 1b, all $p < 0.05$), although there was no difference between the first two groups (Fig. 1b, $p > 0.05$). Nevertheless intramuscular administration of hiPSC-MSCs in the MSC-Saline group achieved a better beneficial effect than intravenous administration of hiPSC-MSCs in the Saline-MSC/once, Saline-MSC/week and Saline-MSC/3 days groups from day 21 onwards (Fig. 1b, all $p < 0.05$).

Taken together, our results demonstrated that systemic intravenous administration of hiPSC-MSCs without intramuscular administration of hiPSC-MSCs improved blood perfusion. Repeated intravenous administration of hiPSC-MSCs every week or every 3 days without intramuscular administration of hiPSC-MSCs further increased blood perfusion compared with a single intravenous injection, although there was no significant difference between intravenous administration repeated every week versus every 3 days. Nonetheless intramuscular administration of hiPSC-MSCs achieved a better beneficial effect than intravenous administration of hiPSC-MSCs once, every week or every 3 days.

### Animal groups and treatment.
Five groups of ICR mice were employed in the main experiment (Fig. 2): (1) ischemia group receiving intravenous administration of saline immediately after induction of ischemia and intramuscular administration of culture medium at day 7; (2) MSC-Saline group receiving intravenous administration of saline immediately after induction of ischemia and intramuscular administration of $3 \times 10^6$ hiPSC-MSCs at day 7; (3) MSC-MSC/once group receiving intravenous administration of $5 \times 10^5$ hiPSC-MSCs immediately after induction of ischemia and intramuscular administration of $3 \times 10^6$ hiPSC-MSCs at day 7; (4) MSC-MSC/week group receiving repeated intravenous administration of $5 \times 10^5$ hiPSC-MSCs immediately and every week following induction of ischemia for 4 weeks and intramuscular administration of $3 \times 10^6$ hiPSC-MSCs at day 7; (5) MSC-MSC/3 days group receiving repeated intravenous administration of $5 \times 10^5$ hiPSC-MSCs immediately and every 3 days following induction of ischemia for 4 weeks and intramuscular administration of $3 \times 10^6$ hiPSC-MSCs at day 7.

### Improved blood perfusion in ischemic hind limb.
Serial laser doppler imaging and analysis was performed to evaluate the blood perfusion and monitor the blood flow recovery in the ischemic hind limb (Fig. 3a). After induction of ischemia, blood perfusion of the ligated limb significantly decreased to an extremely low level relative to the non-ligated limb in the ischemia group ($2.98 \pm 0.56$), MSC-Saline group ($2.96 \pm 0.30$), MSC-MSC/once group ($2.95 \pm 0.48$), MSC-MSC/week group ($3.01 \pm 0.29$) and MSC-MSC/3 days group ($2.97 \pm 0.30$). There was no significant difference between the five groups (Fig. 3b, all $p > 0.05$). These results confirmed that acute hind-limb ischemia was induced in all groups. Intramuscular administration of hiPSC-MSCs with intravenous administration of saline or with intravenous administration of hiPSC-MSCs once or every week or every 3 days in the MSC-Saline, MSC-MSC/once, MSC-MSC/week and MSC-MSC/3 days groups resulted in a significant and progressive improvement in the blood perfusion of the ligated

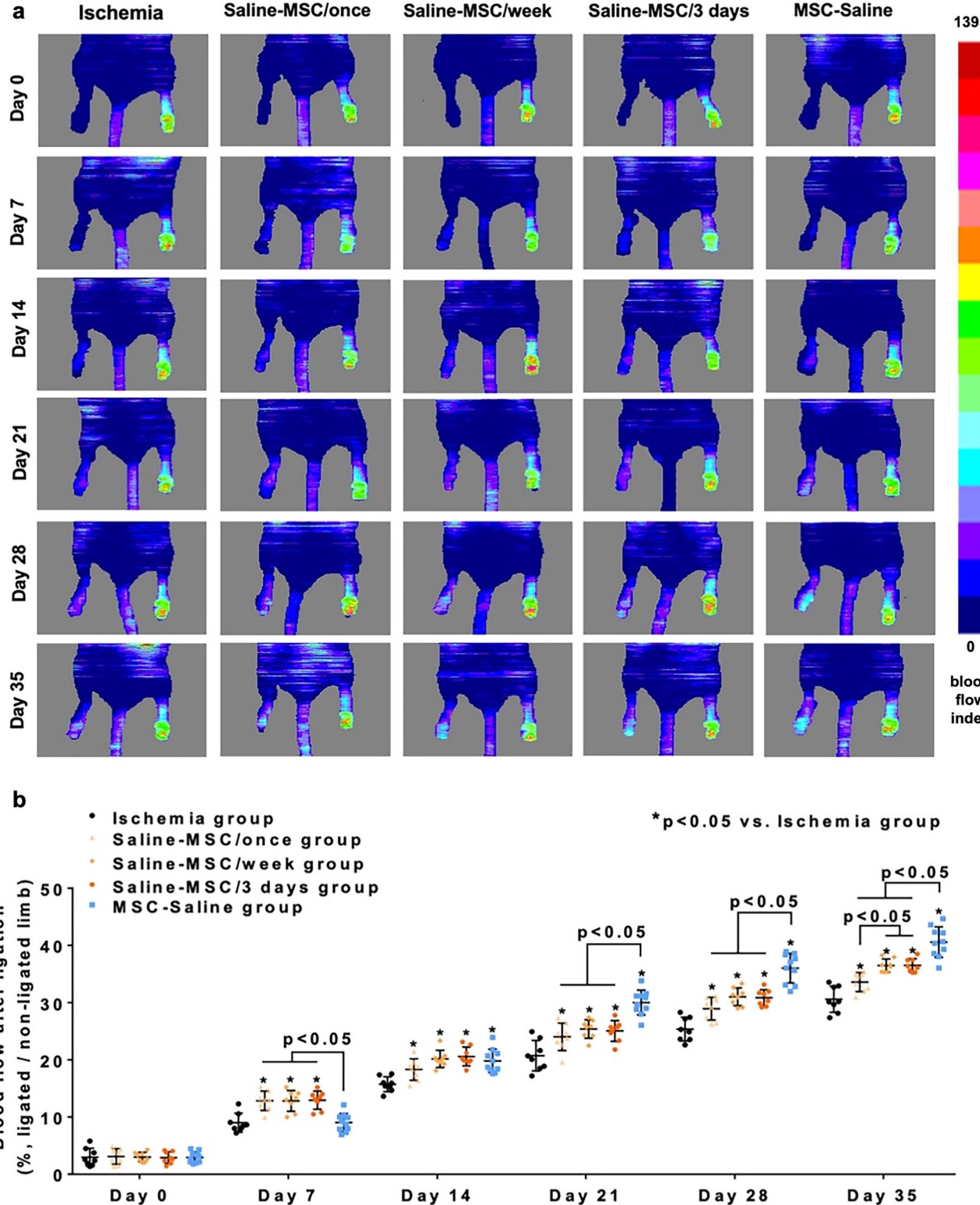

**Fig. 1 Comparison of intravenous administration of hiPSC-MSCs versus intramuscular hiPSC-MSCs delivery.** To evaluate blood perfusion in the groups that received intravenous hiPSC-MSCs infusion without intramuscular hiPSC-MSCs transplantation, Laser Doppler imaging analysis was performed immediately and every week following femoral artery ligation (**a**). A single or repeated intravenous administration of hiPSC-MSCs in the Saline-MSC/once, Saline-MSC/week or Saline-MSC/3 days groups significantly increased blood perfusion from day 7 onwards compared with the ischemia group. Moreover, repeated intravenous hiPSC-MSCs infusion further improved blood perfusion at day 35. Nonetheless intramuscular hiPSC-MSC transplantation in the MSC-Saline group showed a superior beneficial effect over repeated intravenous hiPSC-MSC infusion in the Saline-MSC/week and Saline-MSC/3 days groups (**b**).

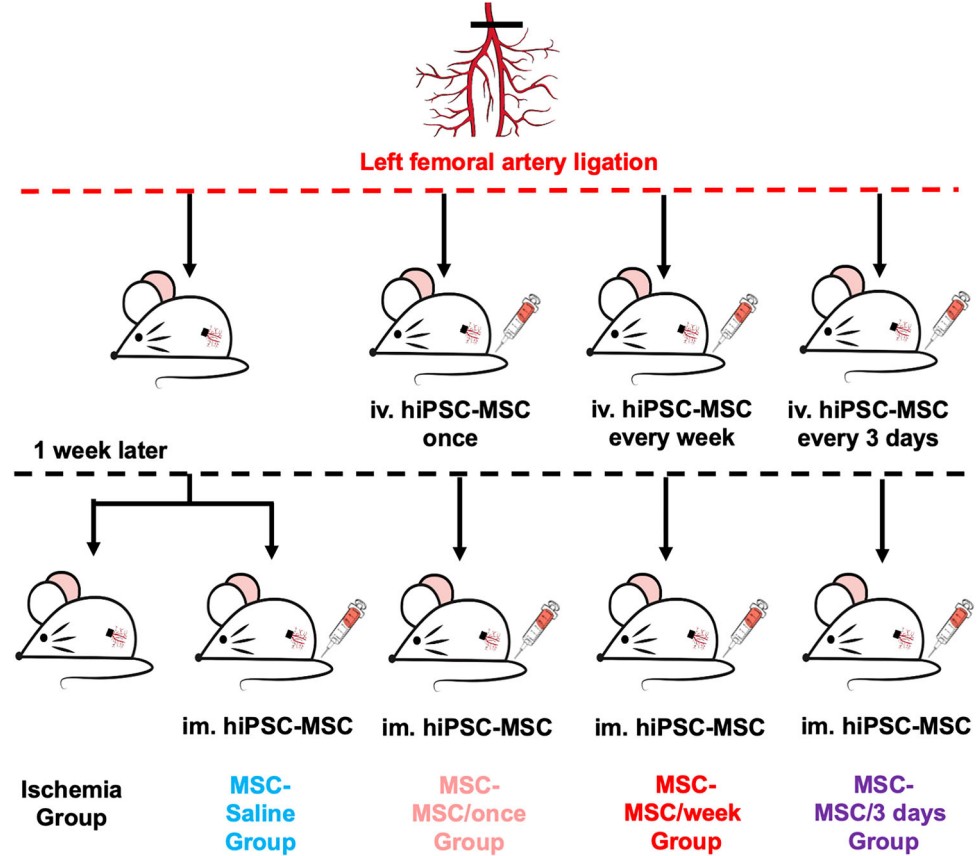

**Fig. 2 Flow chart of the experiment.** There are five groups of ICR mice in main experiment: ischemia group, MSC-Saline group, MSC-MSC/once group, MSC-MSC/week group, MSC-MSC/3 days group.

limb from day 14 onwards compared with the ischemia group (Fig. 3b, all $p < 0.05$). Intravenous administration of hiPSC-MSCs significantly increased blood perfusion in the MSC-MSC/once, MSC-MSC/week and MSC-MSC/3 days groups from day 7 onwards compared with the ischemia and MSC-Saline groups (Fig. 3b, all $p < 0.05$). Repeated intravenous administration of hiPSC-MSCs in the MSC-MSC/week and MSC-MSC/3 days groups further increased blood perfusion from day 28 onwards compared with the MSC-MSC/once group (Fig. 3b, all $p < 0.05$). Nevertheless there was no significant difference between mice that received repeated intravenous administration of hiPSC-MSCs in the MSC-MSC/week versus MSC-MSC/3 days groups throughout the study period. On day 35, blood perfusion of the ligated hind limb in the ischemia, MSC-Saline, MSC-MSC/once, MSC-MSC/ week and MSC-MSC/3 days groups were 30.57 ± 0.81, 40.56 ± 0.84, 44.99 ± 0.75, 50.41 ± 0.68 and 51.12 ± 0.86 respectively.

Taken together, our results showed that systemic intravenous administration of hiPSC-MSCs combined with intramuscular transplantation of hiPSC-MSCs improved blood perfusion in a mouse model of hind-limb ischemia relative to intramuscular hiPSC-MSC transplantation without systemic hiPSC-MSC delivery. In addition, repeated intravenous administration of hiPSC-MSCs every week or every 3 days further improved the therapeutic effects of hiPSC-MSC-based therapy compared with a single intravenous injection. No significant difference was observed between repeated intravenous administration of hiPSC-MSCs every week and every 3 days.

**Increased neovascularization and decreased fibrosis in ischemic hind limb.** To evaluate neovascularization in the ischemic limb, immunohistochemical staining with anti-mouse

alpha-smooth muscle antigen (α-SMA) and anti-mouse von Willebrand factor (vWF) antibodies were performed to assess arteriogenesis and angiogenesis following cellular transplantation respectively (Fig. 4a). On day 14, intramuscular transplantation of hiPSC-MSCs in the MSC-Saline group did not increase arteriogenesis and capillary formation (Fig. 4b, c, $p > 0.05$). Nevertheless, systemic intravenous administration of hiPSC-MSCs in the MSC-MSC/once, MSC-MSC/week and MSC-MSC/3 days groups significantly improved arteriogenesis and capillary formation compared with the ischemia group (Fig. 4b, c, all $p < 0.05$). On day 35, compared with the ischemia group, intramuscular transplantation of hiPSC-MSCs in the MSC-Saline, MSC-MSC/once, MSC-MSC/week and MSC-MSC/3 days groups significantly increased neovascularization (Fig. 4b, c, all $p < 0.05$). Moreover, systemic intravenous administration of hiPSC-MSCs in the MSC-MSC/once, MSC-MSC/week and MSC-MSC/3 days groups further improved neovascularization compared with the MSC-Saline group on day 35 (Fig. 4b, c, $p < 0.05$). In addition, repeated intravenous administration of hiPSC-MSCs in the MSC-MSC/week and MSC-MSC/3 days groups further promoted neovascularization compared with the MSC-MSC/once group (Fig. 4b, c, all $p < 0.05$). There was no difference in neovascularization between the MSC-MSC/week and MSC-MSC/3 days groups (Fig. 4b, c, all $p > 0.05$).

To assess the degree of fibrosis in the ischemic limb, Masson's Trichrome staining were performed to determine the percentage of fibrotic tissue in the ischemic limb (Fig. 4a). On day 14, intramuscular transplantation of hiPSC-MSCs in the MSC-Saline group did not decrease fibrosis (Fig. 4d, $p > 0.05$). Nevertheless, systemic intravenous administration of hiPSC-MSCs in the MSC-MSC/once, MSC-MSC/week and MSC-MSC/3 days groups

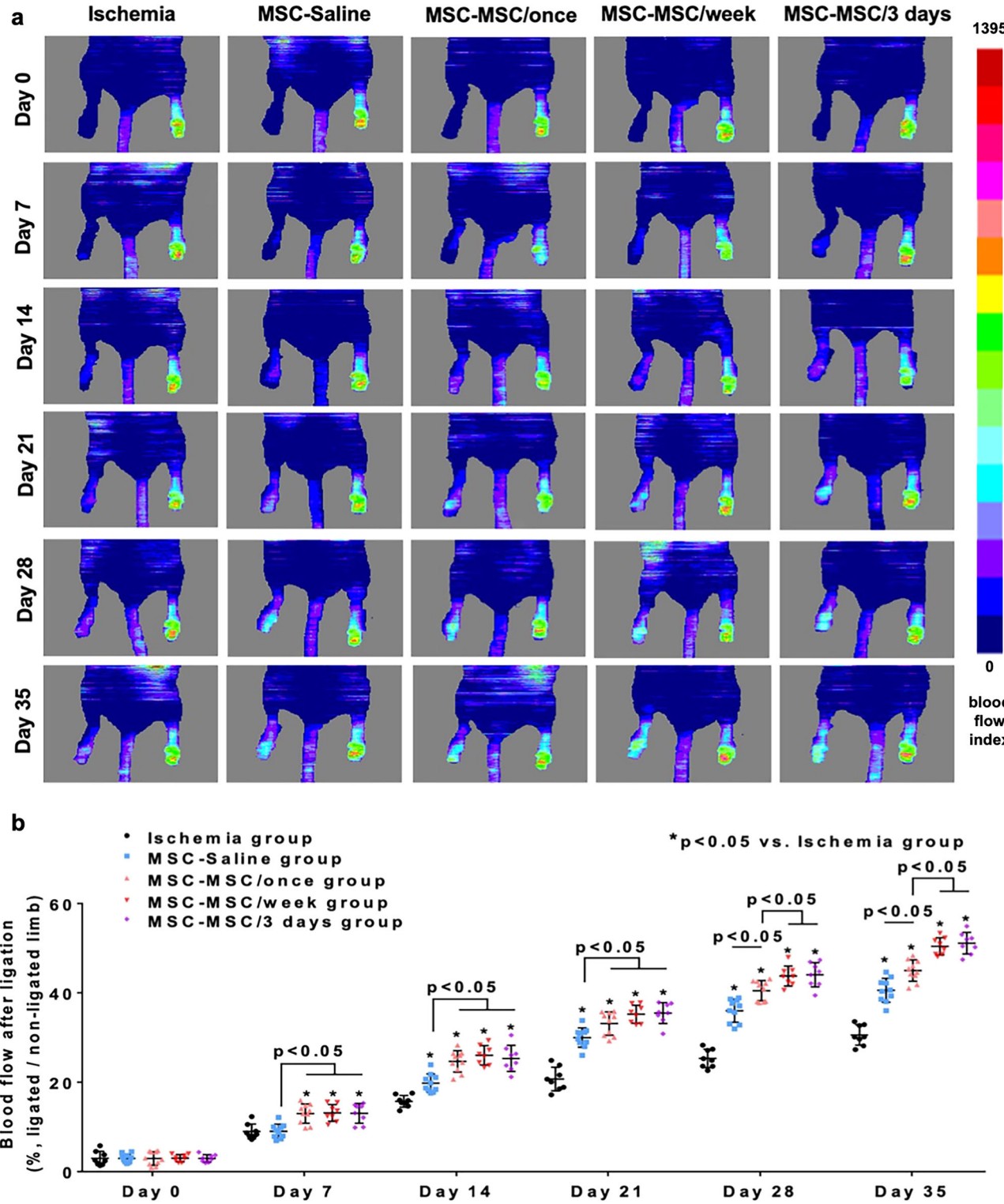

**Fig. 3 Therapeutic efficacy of intravenous cellular administration combined with intramuscular cellular delivery.** Laser Doppler imaging analysis was performed immediately and every week following femoral artery ligation to evaluate blood perfusion in the ischemic hind limbs (**a**). After intramuscular transplantation of hiPSC-MSCs, blood perfusion was significantly improved in the MSC-Saline, MSC-MSC/once, MSC-MSC/week and MSC-MSC/3 days groups compared with the ischemia group from day 14 onwards (all $p < 0.05$). A single and repeated intravenous hiPSC-MSC infusion further improved blood perfusion in the MSC-MSC/once, MSC-MSC/week and MSC-MSC/3 days groups compared with MSC-Saline group (all $p < 0.05$). Moreover, the blood perfusion was significantly higher in the MSC-MSC/week and MSC-MSC/3 days groups compared with the MSC-MSC/once group (all $p < 0.05$). There was no significant difference between the MSC-MSC/week and MSC-MSC/3 days groups ($p > 0.05$) (**b**).

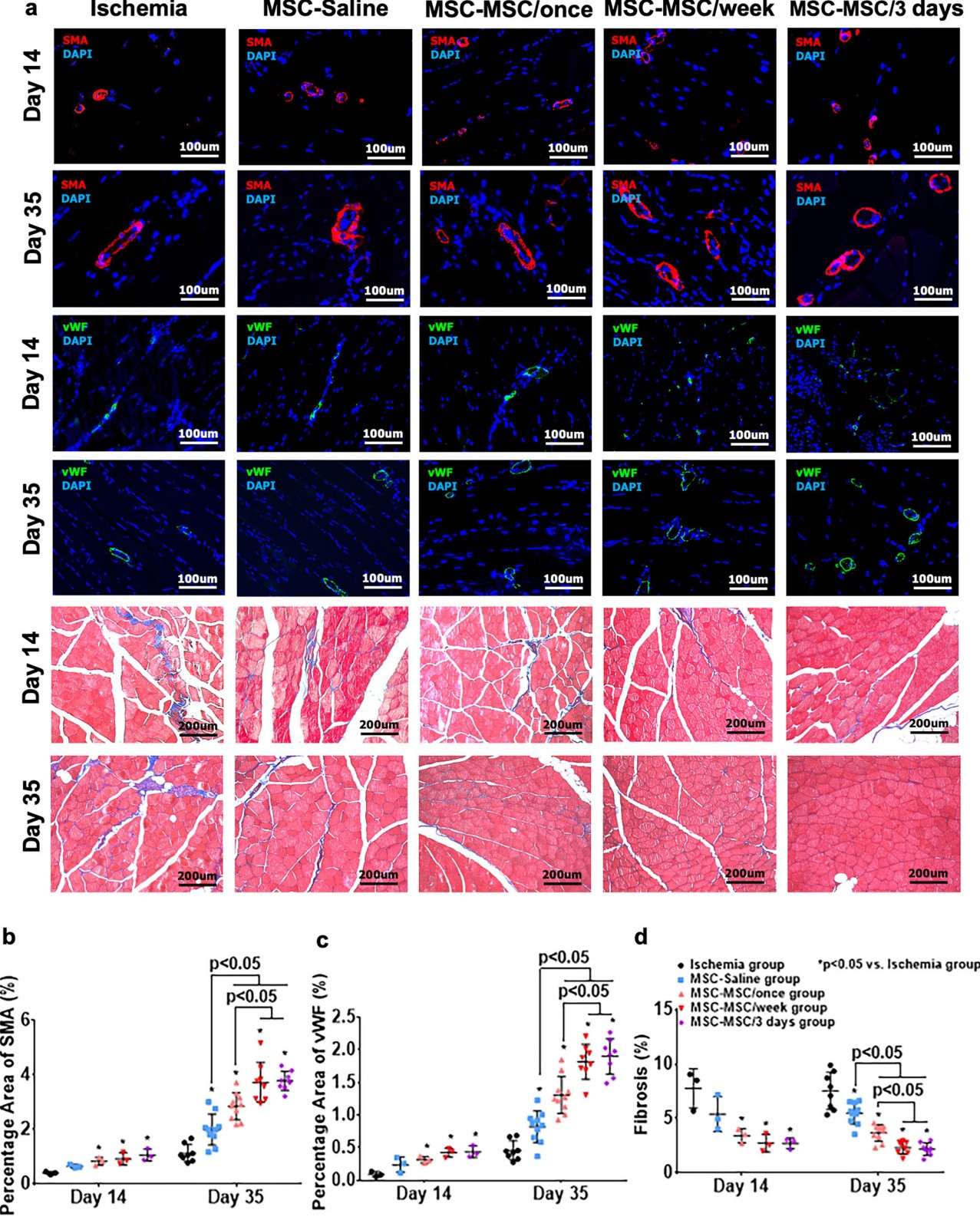

significantly reduced fibrosis compared with the ischemia group (Fig. 4d, all $p < 0.05$). Compared with the ischemia group, intramuscular transplantation of hiPSC-MSCs in the MSC-Saline, MSC-MSC/once, MSC-MSC/week and MSC-MSC/3 days groups significantly ameliorated fibrosis on day 35 (Fig. 4d, all $p < 0.05$). Moreover, systemic intravenous administration of hiPSC-MSCs in the MSC-MSC/once, MSC-MSC/week and MSC-MSC/3 days

groups significantly reduced fibrosis compared with the MSC-Saline group (Fig. 4d, all $p < 0.05$). In addition, repeated intravenous administration of hiPSC-MSCs in the MSC-MSC/week and MSC-MSC/3 days groups further decreased fibrosis compared with the MSC-MSC/once group (Fig. 4d, all $p < 0.05$). There were no differences in fibrosis between the MSC-MSC/week and MSC-MSC/3 days groups (Fig. 4d, all $p > 0.05$).

**Fig. 4 Repeated intravenous administration of hiPSC-MSCs increased neovascularization and decreased fibrosis.** Immunohistochemical staining with anti-mouse vWF (green) and anti-mouse α-SMA (red) antibodies was performed to assess angiogenesis and arteriogenesis in ischemic tissues. Masson's trichrome staining was performed to evaluate the degree of fibrosis (**a**). On day 14, neovascularization was markedly increased in the MSC-MSC/once, MSC-MSC/week and MSC-MSC/3 days groups, not in the MSC-Saline group, relative to the ischemia group. On day 35, after intramuscular transplantation of hiPSC-MSCs, neovascularization was significantly improved in the MSC-Saline, MSC-MSC/once, MSC-MSC/week and MSC-MSC/ 3 days groups compared with the ischemia group (all $p < 0.05$). Intravenous administration of hiPSC-MSCs in the MSC-MSC/once, MSC-MSC/week and MSC-MSC/3 days groups enhanced the therapeutic effects of intramuscularly transplanted hiPSC-MSCs on neovascularization compared with the MSC-Saline group (all $p < 0.05$). Moreover, neovascularization was further enhanced by repeated intravenous hiPSC-MSC infusion in the MSC-MSC/week and MSC-MSC/3 days groups compared with the MSC-MSC/once group (**b, c**). On day 14, fibrosis was remarkably decreased in the MSC-MSC/once, MSC-MSC/week and MSC-MSC/3 days groups, not in the MSC-Saline group, relative to the ischemia group. On day 35, after intramuscular transplantation of hiPSC-MSCs, fibrosis was significantly reduced in the MSC-Saline, MSC-MSC/once, MSC-MSC/week and MSC-MSC/3 days groups compared with the ischemia group (all $p < 0.05$). Intravenous administration of hiPSC-MSCs in the MSC-MSC/once, MSC-MSC/week and MSC-MSC/3 days groups enhanced the therapeutic effects of intramuscularly transplanted hiPSC-MSCs on reduction of fibrosis compared with the MSC-Saline group (all $p < 0.05$). Moreover, the anti-fibrotic effect was further enhanced by repeated intravenous hiPSC-MSC infusion in the MSC-MSC/week and MSC-MSC/3 days groups compared with the MSC-MSC/once group (**d**).

Taken together, our results showed that systemic intravenous administration of hiPSC-MSCs combined with intramuscular transplantation of hiPSC-MSCs promoted neovascularization and reduced fibrosis in a mouse model of hind-limb ischemia. Repeated intravenous administration of hiPSC-MSCs every week or every 3 days further increased the neovascularization and decreased the fibrosis following cellular transplantation compared with a single intravenous injection. No significant difference was observed between repeated intravenous administration of hiPSC-MSCs every week and every 3 days.

**Improved cellular engraftment and survival after transplantation.** Fluorescent imaging of ischemic hind limbs was performed immediately and every week after induction of ischemia to access the cellular engraftment and survival of intramuscularly transplanted hiPSC-MSCs (Fig. 5a). To avoid any confusion on the fluorescent signal, intravenous administered hiPSC-MSCs were not labeled with DiR. There was no significant difference in fluorescent signal intensity over the ischemic hind limb after intramuscular cellular transplantation (Fig. 5b, all $p > 0.05$). Systemic intravenous administration of hiPSC-MSCs significantly increased cellular engraftment and survival in the MSC-MSC/ once, MSC-MSC/week and MSC-MSC/3 days groups from day 14 onwards relative to the MSC-Saline group (Fig. 5b, all $p < 0.05$). Moreover, repeated intravenous administration of hiPSC-MSCs in the MSC-MSC/week and MSC-MSC/3 days groups further improved cellular engraftment and survival from day 21 onwards compared with the MSC-MSC/once group (Fig. 5b, all $p < 0.05$). There was no significant difference between mice that received repeated intravenous administration of hiPSC-MSCs in the MSC/ week and MSC-MSC/3 days groups throughout the study period. On day 35, the estimated survival rates in MSC-Saline, MSC-MSC/once, MSC-MSC/week and MSC-MSC/3 days groups decreased to $2.59 \pm 0.31\%$, $8.33 \pm 0.54\%$, $13.56 \pm 0.49\%$ and $14.23 \pm 0.42\%$, respectively (Supplementary Fig. 2 and Supplementary Data 1).

Cellular engraftment and survival of intramuscularly transplanted hiPSC-MSCs were further confirmed by immunohistochemical double staining with anti-human GAPDH and anti-human mitochondria antibodies (Fig. 6a). Systemic intravenous administration of hiPSC-MSCs significantly increased human GAPDH and human mitochondria positive cells over the ischemic hind limb in the MSC-MSC/once, MSC-MSC/week and MSC-MSC/3 days groups from day 14 onwards relative to the MSC-Saline group (Fig. 6b, all $p < 0.05$). Moreover, on day 35, repeated intravenous administration of hiPSC-MSCs in the MSC-MSC/week and MSC-MSC/3 days groups further increased the human GAPDH and human mitochondria positive cells

compared with the MSC-MSC/once group (Fig. 6b, all $p < 0.05$). No difference between the MSC-MSC/week and MSC-MSC/ 3 days groups was noted (Fig. 6b, all $p > 0.05$).

Taken together, our results demonstrated that systemic intravenous administration of hiPSC-MSCs enhanced engraftment and survival of intramuscularly transplanted hiPSC-MSCs. In addition, repeated intravenous administration every week or every 3 days further increased the cellular engraftment and survival compared with a single intravenous injection. No significant difference was observed between repeated intravenous administration of hiPSC-MSCs every week versus every 3 days.

**Muscular macrophages infiltration and polarization.** Immunohistochemical staining with anti-mouse CD68 antibody was performed to calculate the number of macrophages after cellular transplantation and evaluate the infiltration of macrophages (Fig. 7a). M2 macrophages were further characterized by immunohistochemical staining with anti-mouse Arginase-1 antibody (Fig. 7a). Although there was no significant difference between any of the five groups at day 7 and 14 after induction of ischemia (Fig. 7b, all $p > 0.05$), intramuscular administration of hiPSC-MSCs in the MSC-Saline, MSC-MSC/once, MSC-MSC/ week and MSC-MSC/3 days groups significantly increased M2 macrophage polarization in the ligated limb from day 14 onwards relative to the ischemia group (Fig. 7c, all $p < 0.05$). Moreover, intravenous administration of hiPSC-MSCs remarkedly promoted M2 macrophage polarization in the MSC-MSC/once, MSC-MSC/week and MSC-MSC/3 days groups from day 7 onwards compared with the ischemia and MSC-Saline groups (Fig. 7c, all $p < 0.05$). On day 35, intramuscular administration of hiPSC-MSCs in MSC-Saline group had significantly decreased the infiltration of macrophages although the M2 macrophage percentage was similar to that in the ischemia group (Fig. 7b, c, all $p < 0.05$). Systemic intravenous administration of hiPSC-MSCs in the MSC-MSC/once, MSC-MSC/week and MSC-MSC/3 days groups significantly decreased macrophage infiltration and increased M2 macrophage polarization relative to the MSC-Saline group (Fig. 7b, c, all $p < 0.05$). Repeated intravenous administration of hiPSC-MSCs in the MSC-MSC/week and MSC-MSC/ 3 days groups further reduced the infiltration of macrophages and increased the polarization of M2 macrophages compared with the MSC-MSC/once group (Fig. 7b, c, all $p < 0.05$). There was no noticeable difference in either the infiltration of macrophages or polarization of M2 macrophages between the MSC-MSC/week and MSC-MSC/3 days groups (Fig. 7b, c, all $p > 0.05$).

Taken together, our results demonstrated that systemic intravenous administration of hiPSC-MSCs decreased the infiltration of macrophages and increased the polarization of

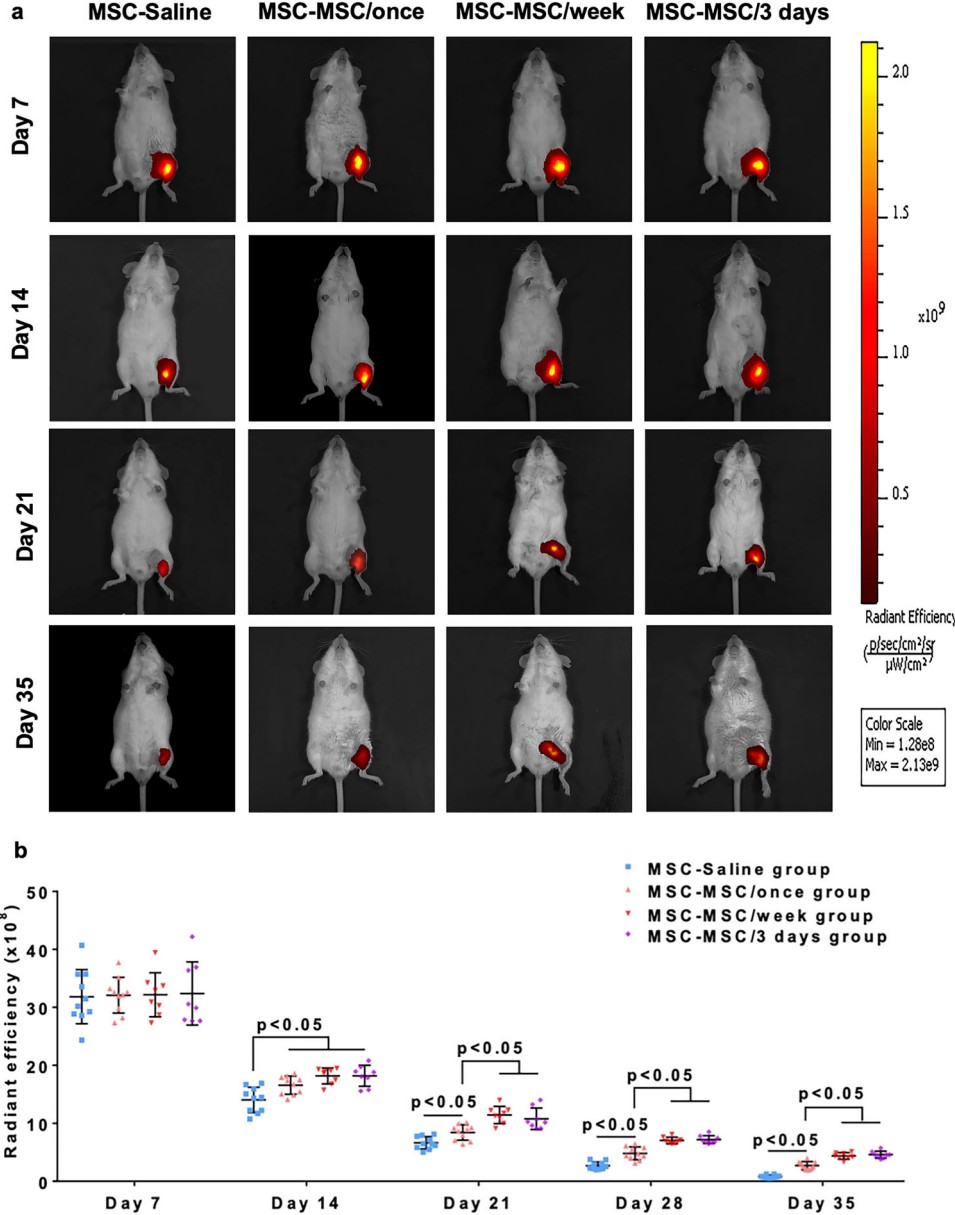

**Fig. 5 Repeated intravenous administration of hiPSC-MSCs prolonged the survival of transplanted hiPSC-MSCs.** A series of fluorescent images of ischemic hind limbs was performed immediately and every week following intramuscular transplantation of hiPSC-MSCs to detect the fate of intramuscularly transplanted hiPSC-MSCs (**a**). A single or repeated intravenous hiPSC-MSCs infusion in the MSC-MSC/once, MSC-MSC/week and MSC-MSC/3 days groups significantly prolonged the survival of intramuscular transplanted hiPSC-MSCs from day 14 onwards compared with the MSC-Saline group (all $p < 0.05$). Moreover, repeated intravenous hiPSC-MSCs infusion in the MSC-MSC/week and MSC-MSC/3 days groups further improved the survival of intramuscularly transplanted hiPSC-MSCs from day 21 onwards compared with the MSC-MSC/once group (all $p < 0.05$), whereas no significant difference was observed between MSC-MSC/week and MSC-MSC/3 days groups ($p > 0.05$) (**b**).

M2 macrophages. Repeated intravenous administration of hiPSC-MSCs every week or every 3 days further decreased the infiltration of macrophages and increased the polarization of M2 macrophages compared with a single intravenous injection, whereas no significant difference was observed between repeated intravenous administration of hiPSC-MSCs every week and every 3 days.

**Increased anti-inflammatory cytokines and decreased inflammatory cytokines.** The limb tissue level of a specific subset-related cytokines was measured using a commercial mouse inflammatory factor array. For anti-inflammatory cytokines, on day 14, there was no significant difference on interleukin (IL)−10

and vascular endothelial growth factor (VEGF) among the ischemia, MSC-Saline and MSC-MSC/once groups (Supplementary Fig. 3a, b, all $p > 0.05$). Nonetheless, repeated systemic intravenous hiPSC-MSC infusion in the MSC-MSC/week and MSC-MSC/3 days groups significantly increased IL-10 and VEGF compared with the ischemia group (Supplementary Fig. 3a, b, all $p < 0.05$). Moreover, an increase of IL-10 was observed in the MSC-MSC/week and MSC-MSC/3 days groups relative to the MSC-Saline group (Supplementary Fig. 3a, b, all $p < 0.05$). On day 35, intramuscular transplantation of hiPSC-MSCs in the MSC-Saline group did not significantly improved IL-10 relative to ischemia group. Nevertheless, systemic intravenous hiPSC-MSC infusion in the MSC-MSC/once, MSC-MSC/week and MSC-

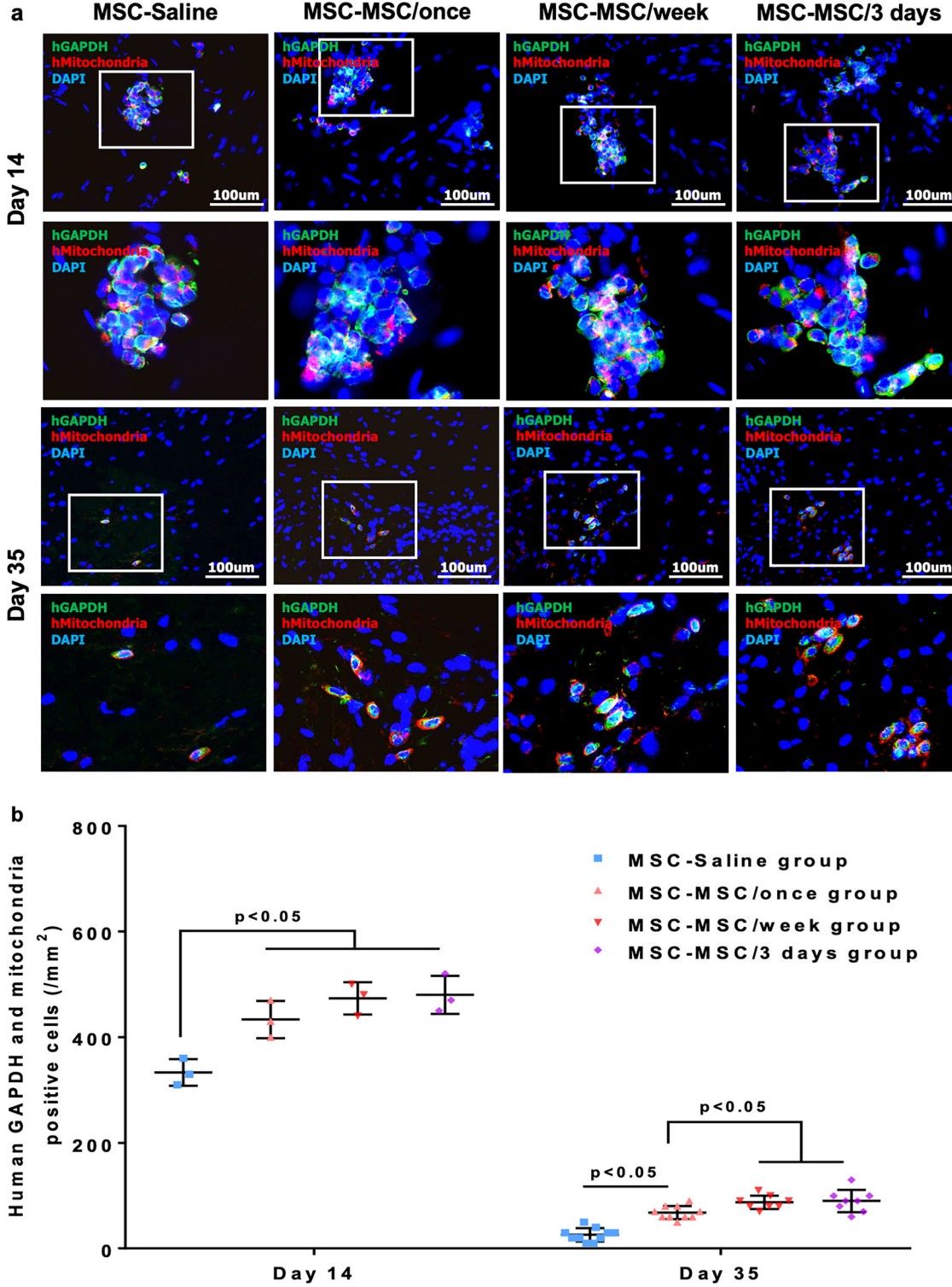

**Fig. 6 Repeated intravenous administration of hiPSC-MSCs improved the engraftment of transplanted hiPSC-MSCs in ischemic tissue.** The engraftment of intramuscularly transplanted hiPSC-MSCs was further confirmed by double immunohistochemical staining with anti-human GAPDH (green) and anti-human mitochondria antibodies (red) at day 14 and 35 (**a**). A single or repeated intravenous hiPSC-MSC infusion in the MSC-MSC/once, MSC-MSC/week and MSC-MSC/3 days groups significantly improved the engraftment of intramuscularly transplanted hiPSC-MSCs from day 14 onwards (all $p < 0.05$). Repeated intravenous hiPSC-MSC infusion in the MSC-MSC/week and MSC-MSC/3 days groups further improved the engraftment of intramuscular transplanted hiPSC-MSCs at day 35 compared with the MSC-MSC/once group (all $p < 0.05$), whereas no significant difference was observed between the MSC-MSC/week and MSC-MSC/3 days groups ($p > 0.05$) (**b**).

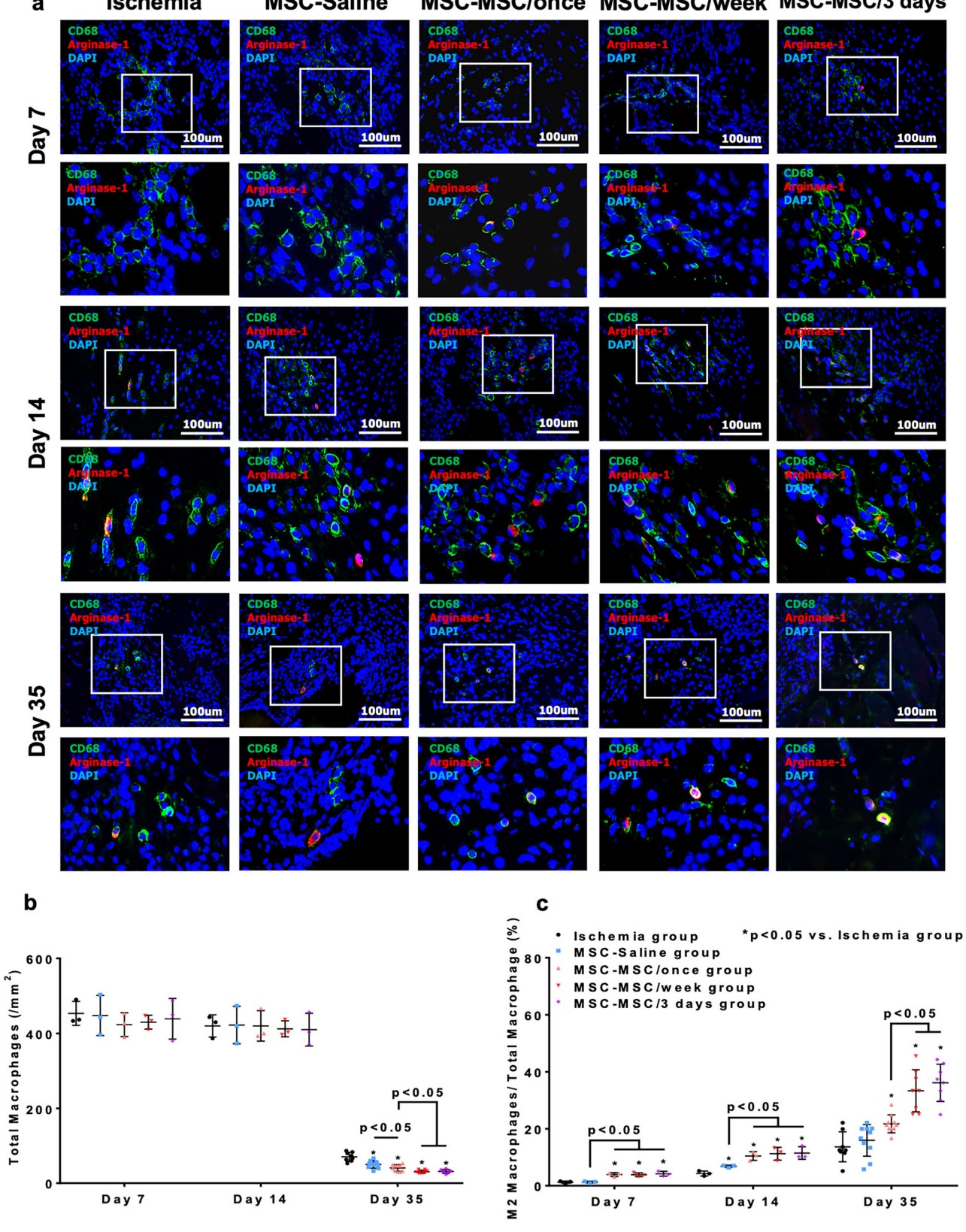

MSC/3 days groups significantly improved IL-10 compared with the ischemia group (Supplementary Fig. 3a, all $p < 0.05$). Moreover, repeated systemic intravenous hiPSC-MSC infusion in the MSC-MSC/week and MSC-MSC/3 days groups further increased IL-10 compared with the MSC-MSC/once group (Supplementary Fig. 3a, all $p < 0.05$). No significant difference on VEGF was

observed among all five groups on day 35 (Supplementary Fig. 3b, all $p < 0.05$).

For inflammatory cytokines, on day 14, intramuscular transplantation of hiPSC-MSCs in the MSC-Saline, MSC-MSC/once, MSC-MSC/week and MSC-MSC/3 days groups significantly decreased IL-1A and IL-17A compared with the ischemia group

**Fig. 7 Repeated intravenous administration of hiPSC-MSCs reduced the infiltration of macrophages and promoted polarization of M2 macrophages.**
Muscular infiltration of macrophages was determined by immunohistochemical staining with anti-mouse CD68 antibody (green) at day 7, 14, and 35. Number of M2 macrophages was detected by immunohistochemical staining with anti-mouse Arginase-1 antibodies (red) (**a**). At day 35, after intramuscular transplantation of hiPSC-MSCs, total macrophages were significantly decreased in the MSC-Saline, MSC-MSC/once, MSC-MSC/week and MSC-MSC/3 days groups compared with the ischemia group (all $p < 0.05$). A single or repeated intravenous hiPSC-MSCs infusion in the MSC-MSC/once, MSC-MSC/week and MSC-MSC/3 days groups significantly decreased the muscular infiltration of macrophages compared with the MSC-Saline group (all $p < 0.05$). In addition, repeated intravenous hiPSC-MSCs infusion in the MSC-MSC/week and MSC-MSC/3 days groups further decreased the muscular infiltration of macrophages compared with the MSC-MSC/once group (all $p < 0.05$). Nevertheless no significant difference was observed between groups at day 7 and 14 (all $p > 0.05$) (**b**). Intramuscular transplantation of hiPSC-MSCs without intravenous hiPSC-MSC infusion significantly increased the polarization of M2 macrophages at day 14 compared with the ischemia group ($p < 0.05$). A single or repeated intravenous hiPSC-MSC infusion in the MSC-MSC/once, MSC-MSC/week and MSC-MSC/3 days groups significantly improved the polarization of M2 macrophages from day 7 onwards (all $p < 0.05$). Repeated hiPSC-MSCs infusion further promoted the polarization of M2 macrophages compared with a single intravenous hiPSC-MSCs infusion in the MSC-MSC/once group at day 35 (all $p < 0.05$) (**c**).

(Supplementary Fig. 3c, d, all $p < 0.05$). Nonetheless, there was no significant difference among the MSC-Saline, MSC-MSC/once, MSC-MSC/week and MSC-MSC/3 days groups (Supplementary Fig. 3c, d, all $p > 0.05$). There was no significant difference on IL-2 and macrophage colony-stimulating factor (MCSF) among the ischemia, MSC-Saline and MSC-MSC/once groups (Supplementary Fig. 3e, f, all $p > 0.05$). Nonetheless, repeated systemic intravenous hiPSC-MSC infusion in the MSC-MSC/week and MSC-MSC/3 days groups significantly decreased IL-2 and MCSF compared with the ischemia group (Supplementary Fig. 3e, f, all $p < 0.05$). On day 35, intramuscular transplantation of hiPSC-MSCs in the MSC-Saline, MSC-MSC/once, MSC-MSC/week and MSC-MSC/3 days groups significantly reduced IL-17A relative to ischemia group (Supplementary Fig. 3d, all $p < 0.05$). Moreover, repeated systemic intravenous hiPSC-MSC infusion in the MSC-MSC/week and MSC-MSC/3 days groups further decreased IL-17A compared with the MSC-Saline and MSC-MSC/once groups respectively (Supplementary Fig. 3d, all $p < 0.05$). No significant difference on IL-1A, IL-2 and MCSF was observed among all five groups on day 35 (Supplementary Fig. 3c, e, f, all $p > 0.05$).

Taken together, our results demonstrated that systemic intravenous administration of hiPSC-MSCs could improve anti-inflammatory cytokines and decreased inflammatory cytokines. Repeated intravenous administration of hiPSC-MSCs every week or every 3 days further improved anti-inflammatory cytokines and decreased inflammatory cytokines compared with a single intravenous injection. No significant difference was observed between repeated intravenous administration of hiPSC-MSCs every week and every 3 days.

**Immunomodulatory effect of systemic administration of hiPSC-MSCs.** Flow cytometry analysis of fresh splenocytes was performed to assess splenic Tregs and natural killer (NK) cells populations and so determine the in vivo immunomodulatory effect of systemic administration of hiPSC-MSCs (Fig. 8a). Splenic NK cells were defined as both a CD49b-FITC and NK1.1-APC positive cell population. Our result showed that splenic NK cells progressively decreased following intramuscular hiPSC-MSC transplantation or intravenous hiPSC-MSC infusion in the MSC-Saline, MSC-MSC/once, MSC-MSC/week and MSC-MSC/3 days groups, whereas no significant difference was noted between different time points in the ischemia group (Supplementary Fig. 4a). Compared with the ischemia group, intramuscular administration of hiPSC-MSCs in the MSC-Saline, MSC-MSC/once, MSC-MSC/week and MSC-MSC/3 days groups significantly decreased splenic NK cells from day 14 onwards (Fig. 8b, all $p < 0.05$). Systemic intravenous hiPSC-MSC infusion in the MSC-MSC/once, MSC-MSC/week and MSC-MSC/3 days groups significantly reduced splenic NK cells from day 7 onwards relative to the ischemia and MSC-Saline groups (Fig. 8b, all

$p < 0.05$). Repeated systemic intravenous hiPSC-MSC infusion in the MSC-MSC/week and MSC-MSC/3 days groups further reduced splenic NK cells from day 14 onwards compared with the MSC-MSC/once group (Fig. 8b, all $p < 0.05$). Nonetheless no significant difference was observed between the MSC-MSC/week and MSC-MSC/3 days groups (Fig. 8b, all $p > 0.05$).

Splenic Tregs were determined as Foxp3 positive cells in a proportion of pre-gated CD4 positive cells. Our result showed that splenic Tregs reached a peak on day 7 in the MSC-MSC/once group, whereas these immunomodulatory cells continued to increase in the MSC-MSC/week and MSC-MSC/3 days groups. No significant difference was observed between different time points in the ischemia and MSC-Saline groups (Supplementary Fig. 4b). Compared with the ischemia group, intramuscular administration of hiPSC-MSCs in the MSC-Saline, MSC-MSC/once, MSC-MSC/week and MSC-MSC/3 days groups significantly increased splenic Tregs on day 35 (Fig. 8c, all $p < 0.05$). Intravenous hiPSC-MSC infusion in the MSC-MSC/once, MSC-MSC/week and MSC-MSC/3 days groups significantly improved splenic Tregs from day 7 onwards compared with the ischemia and MSC-Saline groups (Fig. 8c, all $p < 0.05$). Repeated systemic intravenous hiPSC-MSCs infusion in the MSC-MSC/week and MSC-MSC/3 days groups further increased splenic Tregs from day 14 onwards compared with the MSC-MSC/once group (Fig. 8c, all $p < 0.05$), but there was no significant difference between the MSC-MSC/week and MSC-MSC/3 days groups (Fig. 8c, all $p > 0.05$).

Taken together, our results demonstrated that systemic intravenous administration of hiPSC-MSCs could modulate systemic immune cell activation by decreasing splenic NK cells as well as increasing splenic Tregs. Repeated intravenous administration of hiPSC-MSCs every week or every 3 days further decreased splenic NKs and increased splenic Tregs compared with a single intravenous injection. No significant difference was observed between repeated intravenous administration of hiPSC-MSCs every week and every 3 days.

**The comparison of therapeutic efficacy of intravenous hiPSC-MSCs infusion and subcutaneous administration of cyclosporine A.** To compare the survival and engraftment of intramuscularly transplanted hiPSC-MSCs with intervenous infusion of hiPSC-MSCs and subcutaneous administration of cyclosporine A, fluorescent imaging of ischemic hind limb was performed immediately and every week in the MSC-Saline-Cyc, MSC-MSC/once-Cyc and MSC-MSC/week-Cyc groups (Supplementary Fig. 5a). There was no significant difference in cellular engraftment between the MSC-MSC/once and MSC-Saline-Cyc groups through this study (Supplementary Fig. 5b, $p > 0.05$). Although repeated intravenous infusion of hiPSC-MSCs without subcutaneous administration of cyclosporine A remarkably increased

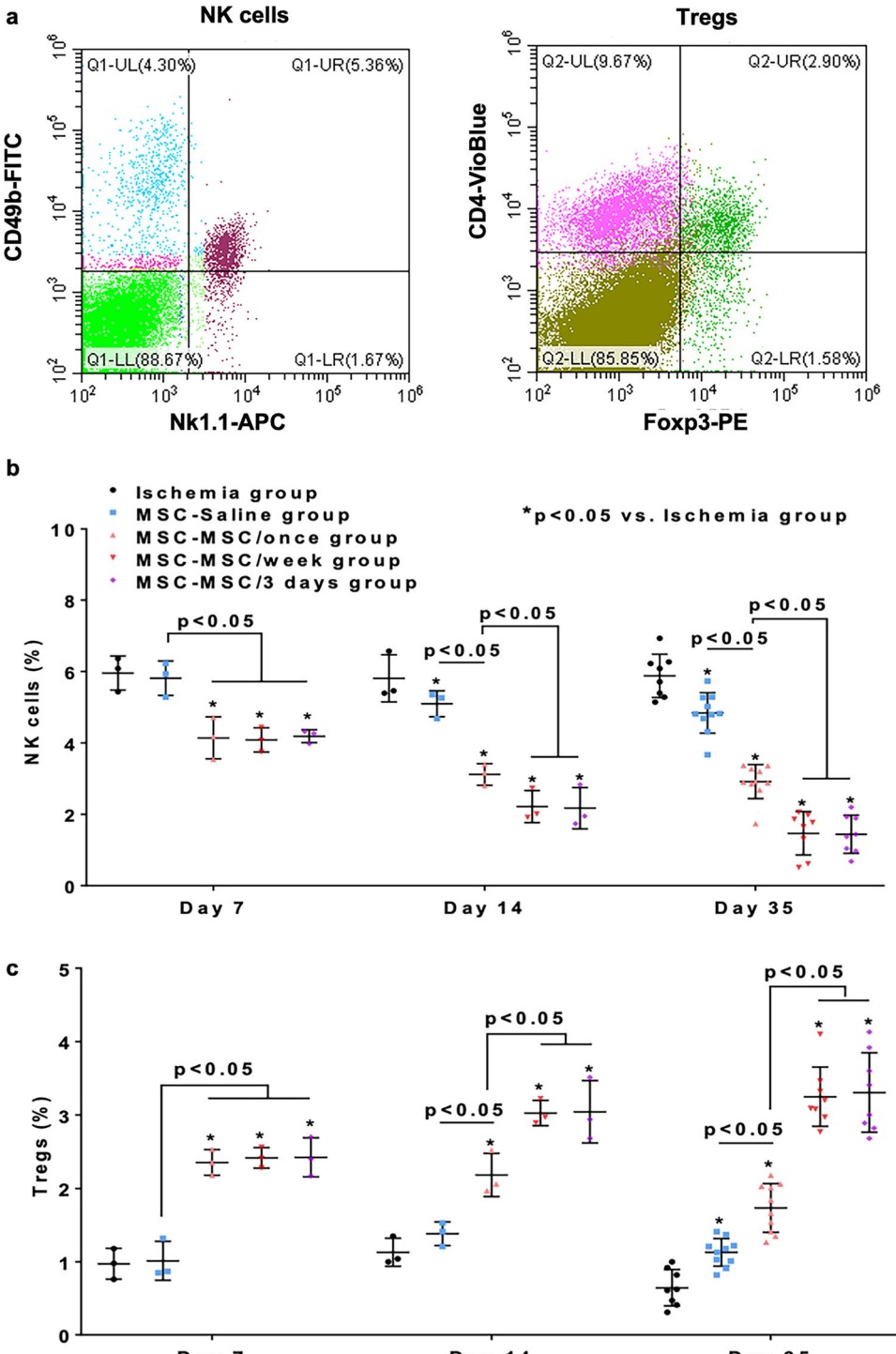

**Fig. 8 Repeated intravenous administration of hiPSC-MSCs reduced splenic NK cells and increased splenic Tregs.** Splenic Tregs and NK cells were determined by flow cytometry analysis at day 7, 14 and 35 (**a**). After intramuscular transplantation of hiPSC-MSCs, splenic NK cells were significantly decreased in the MSC-Saline, MSC-MSC/once, MSC-MSC/week and MSC-MSC/3 days groups from day 14 onwards compared with the ischemia group (all $p < 0.05$). A single or repeated intravenous hiPSC-MSC infusion in the MSC-MSC/once, MSC-MSC/week and MSC-MSC/3 days groups significantly decreased splenic NK cells from day 7 onwards compared with the ischemia and MSC-Saline groups (all $p < 0.05$). Repeated intravenous hiPSC-MSC infusion in the MSC-MSC/week and MSC-MSC/3 days groups further decreased splenic NK cells from day 14 onwards compared with the MSC-MSC/once group (all $p < 0.05$) (**b**). After intramuscular transplantation of hiPSC-MSCs, splenic Tregs were significantly increased in the MSC-Saline, MSC-MSC/once, MSC-MSC/week and MSC-MSC/3 days groups at day 35 compared with the ischemia group (all $p < 0.05$). A single or repeated intravenous hiPSC-MSC infusion in the MSC-MSC/once, MSC-MSC/week and MSC-MSC/3 days groups significantly increased splenic Tregs compared with the ischemia and MSC-Saline groups (all $p < 0.05$). Moreover, repeated intravenous hiPSC-MSC infusion in the MSC-MSC/week and MSC-MSC/3 days groups further increased splenic Tregs from day 14 onwards compared with the MSC-MSC/once group (all $p < 0.05$) (**c**).

cell engraftment in the MSC-MSC/week group relative to the MSC-MSC/once group (Supplementary Fig. 5b, $p < 0.05$), no significant difference was observed after subcutaneous administration of cyclosporine A between the MSC-MSC/week-Cyc and MSC-MSC/once-Cyc groups (Supplementary Fig. 5b, $p > 0.05$). Nonetheless, subcutaneous administration of cyclosporine A did not improve the cell engraftment in the MSC-MSC/once-Cyc and MSC-MSC/week-Cyc groups relative to the MSC-MSC/once and MSC-MSC/week groups respectively (Supplementary Fig. 5b, $p > 0.05$).

To compare the therapeutic efficacy of intramuscularly transplanted hiPSC-MSCs with intervenous infusion of hiPSC-MSCs and subcutaneous administration of cyclosporine A, serial laser doppler imaging and analysis was performed to evaluate the blood perfusion and monitor the blood flow recovery in the ischemic hind limb (Supplementary Fig. 6a). When comparison between the MSC-MSC/once and MSC-Saline-Cyc groups was performed, intravenous infusion of hiPSC-MSCs significantly improved blood perfusion in the MSC-MSC/once group relative to MSC-Saline-Cyc group during the first 2 weeks (Supplementary Fig. 6b, $p < 0.05$). Following intramuscular hiPSC-MSC transplantation at day 7, blood perfusion progressly increased in the MSC-MSC/once and MSC-Saline-Cyc groups. Nevertheless, no significant difference was observed between the MSC-MSC/once and MSC-Saline-Cyc groups from day 21 onwards (Supplementary Fig. 6b, $p > 0.05$). Repeated intravenous infusion of hiPSC-MSCs with or without subcutaneous administration of cyclosporine A significantly improved blood perfusion at day 35 in the MSC-MSC/week and MSC-MSC/week-Cyc groups compared with the MSC-MSC/once and MSC-MSC/once-Cyc groups respectively (Supplementary Fig. 6b, $p < 0.05$). Nonetheless, subcutaneous administration of cyclosporine A did not improve the blood perfusion in the MSC-MSC/once-Cyc and MSC-MSC/week-Cyc groups relative to the MSC-MSC/once and MSC-MSC/week groups respectively (Supplementary Fig. 6b, $p > 0.05$).

Cumulatively, our results demonstrated that no significant difference was observed in cell engraftment between a single or repeated intravenous hiPSC-MSC infusion and subcutaneous administration of cyclosporine A. Although there was no significant difference in blood perfusion between the cyclosporine A and single hiPSC-MSC infusion, a significantly improved blood perfusion was observed in the repeated hiPSC-MSC infusion groups relative to the cyclosporine A group. Furthermore, subcutaneous administration of cyclosporine A did not further increased cell engraftment or therapeutic efficacy in either single or repeated hiPSC-MSC infusion groups.

## Discussion

In this study, we provide novel evidence to the best of our knowledge that a combination of repeated systemic intravenous administration and local transplantation of hiPSC-MSCs can enhance the therapeutic efficacy of MSC-based therapy in a mouse model of hind-limb ischemia. The increased therapeutic efficacy following repeated intravenous administration was due to both increased engraftment of intramuscular transplanted hiPSC-MSCs and immunomodulatory effect of intravenous infused hiPSC-MSCs. Repeated intravenous administration of hiPSC-MSCs did not increase the risk of tumor formation anywhere. Nevertheless, subcutaneous administration of cyclosporine A did not further increased the therapeutic efficacy of MSC-based therapy.

First, our results demonstrated that although the total number of hiPSC-MSCs infused was larger in the Saline-MSC/week group and Saline-MSC/3 days group compared with the MSC-saline group, repeated intravenous administration of hiPSC-MSCs failed

to recapitulate the beneficial effects of intramuscularly transplanted hiPSC-MSCs. These results were supported by the observation that even though intravenously administered hiPSC-MSCs could migrate to the site of injury through "cell homing", cellular engraftment around the injury site was remarkably low compared with that following intramuscular transplantation. Therefore, local delivery of hiPSC-MSCs demonstrated superior efficacy over intravenous administration. Second, a combination of single or repeated intravenous administration and local transplantation of hiPSC-MSCs significantly enhanced the therapeutic efficacy of MSC-based therapy. These improvements were associated with the immunomodulatory effect of intravenous cellular infusion as well as the increased cellular survival and engraftment of intramuscular cellular transplantation. Third, repeated intravenous administration of hiPSC-MSCs elicited a superior therapeutic response than a single injection with or without intramuscular transplantation of hiPSC-MSCs. This finding is consistent with recent studies wherein the therapeutic efficacy of cell-based therapies was enhanced by repeated rather than increased dosing[14–16]. Nevertheless, repeated intravenous administration of hiPSC-MSCs every 3 days did not further improved the therapeutic efficacy compared with repeated hiPSC-MSCs infusion every week. Finally, improved therapeutic efficacy of repeated intravenous administration of hiPSC-MSCs was observed relative to a single intravenous infusion regardless of subcutaneous administration of cyclosporine A or not. Nevertheless, subcutaneous administration of cyclosporine A did not further increased cell engraftment or therapeutic efficacy in either single or repeated hiPSC-MSC infusion groups. In summary, our results support the application of a combination of intramuscular hiPSC-MSC transplantation and repeated intravenous hiPSC-MSC infusion every week without subcutaneous administration of cyclosporine A to improve the therapeutic efficacy of cell-based therapies.

Adult stem cells harvested from aged patients exhibit compromised proliferation and angiogenesis potential that limits their therapeutic effect in cardiovascular disease[4]. Pluripotent stem cell-derived cells with their unlimited cell source, such as hiPSC-MSCs, are a promising candidate for treatment of cardiovascular disease[7,8]. hiPSC-MSCs are not only morphologically similar to BM-MSCs, but also expressing similar antigens including CD44, CD49a, CD49e, CD73, CD105 and CD166[7,8]. In addition, hiPSC-MSCs possess multiple immunomodulatory actions similar to BM-MSCs, suppressing proliferation, cytokine secretion and cytotoxicity of immune T-cells; and modulating the functions of Tregs and NK cells[7,12]. Our previous study showed that hiPSC-MSCs offered an unlimited "off-the-shelf" cell source with predictable therapeutic efficacy. In addition, our recent studies demonstrated that hiPSC-MSCs achieved a superior beneficial effect compared with BM-MSCs on induction of neovascularization due to their high engraftment rate and insensitivity to interferon (IFN)-γ-induced human leukocyte antigen (HLA) expression[7,17]. Nevertheless, immunological barriers have prevented translation of these pre-clinical results to clinical application. Although hiPSC-MSCs express few HLA molecules, the long-term engraftment of transplanted hiPSC-MSCs rapidly declines[7,8]. In our recent study, we observed an improved engraftment of transplanted hiPSC-MSCs or hiPSC-CMs after pre-transplantation systemic intravenous administration of hiPSC-MSCs[12]. This improvement was associated with increased systemic Tregs and decreased circulating NK cells. Nonetheless systemic Tregs tended to decrease after reaching a peak level 7 days following a single intravenous injection. It remains unknown whether the immunomodulatory effects of hiPSC-MSCs can be further enhanced through repeated intravenous administration. In this study, although an increased dose of

intravenously administered hiPSC-MSCs did not elicit an enhanced immunomodulatory response, repeated administration achieved a superior beneficial effect compared with a single dosage. Next, our results showed that systemic intravenous administration of hiPSC-MSCs every week was sufficient to enhance the improved systemic immunomodulatory effect. When the frequency of systemic intravenous cellular infusion increased to every 3 days, no further improvement was observed.

Previous studies demonstrated that MSC was a group of cells releasing a large spectrum of immunomodulatory factors as well as inhibiting a wide repertoire of pro-inflammatory factors[18]. Our result confirmed the immunomodulatory effect of hiPSC-MSCs on cytokine release. In addition, our study further demonstrated that local cell transplantation only modulated a few inflammatory cytokines, such as IL-1A and IL-17A. Nonetheless, intravenous cell delivery has superior immunomodulatory efficacy on inflammatory and anti-inflammatory cytokines. Moreover, the immunomodulatory efficacy could be improved by repeated intravenous dosages. Taken together, our results showed intravenous hiPSC-MSC infusion modulate a wide variety of inflammatory and anti-inflammatory, which could not be achieved by local cell transplantation.

Similar to our previous study, splenic Tregs reached a peak level around 7 days following a single hiPSC-MSC infusion. Number of splenic Tregs continued to increase after each repeated infusion of hiPSC-MSCs. In addition, number of NK cells progressively reduced over the first 2 weeks but thereafter remained stable. Repeated hiPSC-MSC infusion could further decrease NK cells to a lower level. Next, promotion of muscular M2 macrophage polarization was also observed in the groups with intravenous hiPSC-MSC infusion and an increase in the proportion of M2 macrophages was observed in the group with repeated infusions relative to the single infusion group. Our results were consistent with recent observations that major mechanisms of the immunomodulatory effects of intravenously administered hiPSC-MSCs are regulated by increasing numbers of systemic Tregs, decreasing NK cells and promoting anti-inflammatory macrophage polarization[13,19,20]. Taken together, our results provide a proof-of-principle that repeated infusion of hiPSC-MSCs has an enhanced immunomodulatory effect on activation of immune cells compared with a single cellular infusion.

In conclusion, our results provide important proof-of-principle data to support future studies on combination of local cellular transplantation together with repeated intravenous cellular infusion in regenerative therapies for ischemic hind limbs. The immunomodulatory effect of intravenous administration of hiPSC-MSCs contributed to tissue repair as well as engraftment of local transplanted hiPSC-MSCs.

There are several limitations to this study. First, intravenous hiPSC-MSC infusion and intramuscular transplantation was performed in an acute scenario of ischemic cardiovascular disease. It remains unclear whether the enhanced immunomodulatory effects of hiPSC-MSCs can be achieved by repeated cellular infusion and what is the optimal timing and dosage in chronic ischemic cardiovascular disease. Our previous study established a porcine model of ischemic heart failure (HF)[8,21]. It deserves future investigation on the effects of repeated systemic intravenous infusion of hiPSC-MSCs in this large animal model of HF. The optimal strategy of repeated intravenous hiPSC-MSCs infusion also needs to be further verified in a large animal model that more closely mimics clinical practice prior to clinical translation. Second, although hiPSC-MSCs are insensitive to IFN-γ-induced HLA expression, some HLA molecules are expressed on the surface of hiPSC-MSCs[17]. The cell survival still low several weeks after transplantation. Recent achievements in gene-editing

technologies such as CRISPR-Cas9 make it feasible to generate universal cells by HLA engineering[22]. HLA class I molecule-free hiPSCs and hESCs have been generated in previous studies[23,24]. Whether MSCs generated from these HLA class I molecule-free hiPSCs can elicit an decreased immunoreaction and enhanced immunomodulatory effects deserves future investigation. Third, although no tumor formation was detected following intravenous hiPSC-MSC infusion in this study, the heterogeneous differentiation as well as the risk of tumor formation should to be addressed prior to clinical application. Intravenous administration of hiPSC-MSC-derived exosomes instead of hiPSC-MSCs can avoid these problems and also warrants future investigation[25,26]. Fourth, our study focused on investigating the immunomodulatory effect of systemic intravenous hiPSC-MSC infusion. The underlying mechanism why subcutaneous administration of cyclosporine A did not improve cell engraftment in single or repeated intravenous hiPSC-MSC infusion groups remains unknown. Previous study demonstrated that the immunosuppressive effect of immunosuppressive medicines is vary depending on the types and the dosage of the drugs[27]. In this study, we administered cyclosporine A at a dosage of 5 mg/kg. Whether other immunosuppressive drugs or other dosage of cyclosporine A could improve cell engraftment under intravenous hiPSC-MSC infusion remains to be clarified in the future study. Fifth, consistent with previous studies, our study showed ischemia mice were benefited from the paracrine effects of hiPSC-MSC-based therapy. Our study cannot rule out the trans-differentiation and cell fusion of transplanted cells. Whether there is any trans-differentiation and cell fusion of transplanted cells after intramuscular transplantation need to be confirmed in the future experiment.

## Methods

**Ethic statement.** All procedures that involved animals were approved by the Committee on the Use of Live Animals in Teaching and Research (CULATR) at the University of Hong Kong. (CULATR number 4777-18). All procedures conformed to the NIH Guide for the Care and Use of Laboratory Animals.

**Human induced pluripotent stem cells culture differentiation and preparation.** The hiPSC line IMR90-iPSCs used in this study were acquired from WiCell Research Institute (Madison, WI, USA).

**Differentiation and culture of hiPSC-MSC.** We followed an established protocol described in our previous study to generate hiPSC-MSCs[7,8,17]. In brief, iPSC-MSCs were differentiated in knockout Dulbecco modified Eagle's medium (10829018, Thermo Fisher Scientific, NY, USA) with 10% fetal bovine serum (26140079, Thermo Fisher Scientific, NY, USA), supplemented with basic fibroblast growth factor (10 ng/mL, PHG0360, Thermo Fisher Scientific, NY, USA), platelet-derived growth factor AB (PDGFAB) (10 ng/mL, 100-00AB, Peprotech, Rocky Hill, NH, USA) and epidermal growth factor (EGF) (10 ng/mL, AF-100-15, Peprotech, Rocky Hill, NH, USA). After one week, hiPSC-MSCs were purified by sorting for CD105$^+$/CD24$^-$ cells using the MoFlo Cell sorting system (Beckman-coulter, USA). Sorted cells were cultured and expanded in MSC medium (05401, Stemcells Technologies, Toronto, Canada). The purified hiPSC-MSCs exhibited CD44, CD49a&e, CD73, CD105, and CD166 expression but not CD45, CD34, or CD133. The morphology of hiPSC-MSCs was similar to that of bone marrow derived MSCs (BM-MSCs) with osteoblast, adipocyte, and chondroblast differentiation potential. Passage 4–6 hiPSC-MSCs were used for transplantation.

**Cell labeling and in vivo imaging.** Near-infrared dye DiR was used to detect transplanted cells in vivo since it has high tissue penetration and a low false-positive signal[4,5]. Three million hiPSC-MSCs were stained with 5 μg/mL DiR cell-labeling solution (D12731, Life Technologies, Carlsbad, CA, USA) for 15 min at 37 °C, then washed and re-suspended in 30 μl fresh plain medium for intramuscular transplantation[6]. To quantify and detect the biodistribution of transplanted cells, animals were prepared for epi-fluorescent imaging using an IVIS® Spectrum optical imaging system under field of view 3.9 × 3.9 cm (PerkinElmer, Inc. MA, USA). All the animals laid on the back. The engrafted DiR-labeled cells were visualized under a 750 nm/800 nm excitation/emission wavelength with fluorescent intensity represented by radiant efficiency. To estimate the survival rate of the transplanted cells, the averaged radiant efficiency was calculated. The estimated survival rates at day 14, 21, 28, or 35 was calculated as the percentage of the radiant

efficiency at day 14, 21, 28, or 35 versus that of the averaged radiant efficiency just after intravenous transplantation.

**The optimal time and dosage for intravenous hiPSC-MSCs infusion**. The optimal time and dosage of systemic intravenous infusion of hiPSC-MSCs was determined by review of three different time points and dosages. Three groups of ICR mice were employed: (1) a single intravenous administration of $2.5 \times 10^5$ hiPSC-MSCs in mice without induction of ischemia ($2.5 \times 10^5$ group, $n = 9$); (2) a single intravenous administration of $5 \times 10^5$ hiPSC-MSCs in mice without induction of ischemia ($5 \times 10^5$ group, $n = 9$); (3) a single intravenous administration of $7.5 \times 10^5$ hiPSC-MSCs in mice without induction of ischemia ($7.5 \times 10^5$ group, $n = 9$). 3 mice in each group were sacrificed on day 3, 7, and 11 for flow cytometry analysis of splenic Tregs. Splenic Tregs in all three groups reached their peak level around 7 days after a single cellular infusion (Supplementary Fig. 7a). Splenic Tregs were higher in the $5 \times 10^5$ or $7.5 \times 10^5$ groups compared with the $2.5 \times 10^5$ group at day 7 or day 11, respectively (Supplementary Fig. 7a, all $p < 0.05$). However, no significant difference was observed between the $5 \times 10^5$ and $7.5 \times 10^5$ groups at day 3, day 7 or day 11 (Supplementary Fig. 7a, all $p > 0.05$). There is no significant difference in splenic $CD4^+$ T cells within or between all three groups (Supplementary Fig. 7b, all $p > 0.05$). Therefore, $5 \times 10^5$ hiPSC-MSCs were intravenous deliveried in this experiment and intramuscular transplantation was performed 7 day after intravenous cell infusion.

**Animal model and transplantation strategy**. Hind limb ischemia was achieved by direct ligation of the left femoral artery following a standard established protocol[7]. In brief, all mice were first anesthetised by intraperitoneal injection of 100 mg/kg ketamine mixed with 10 mg/kg xylazine. Mice were then secured on a temperature-controlled heating pad and the hair surrounding the surgical site removed by depilatory cream. Next, a 5 mm incision was made along the femoral vessel to explore the femoral artery, vein and nerve. We ligated the femoral artery, taking care to avoid any injury to the vein and nerve. Intravenous injection of either 100 μl saline or $5 \times 10^5$ hiPSC-MSCs prepared in 100 μL normal saline via the tail vein were performed immediately after model induction. Then all ischemic animals were randomized to receive repeated intravenous administration of $5 \times 10^5$ hiPSC-MSCs every week or every 3 days or not at all. One week after induction of ischemia, animals were randomized to receive direct intramuscular injection of culture medium (100 μl), or $3 \times 10^6$ DiR labeled hiPSC-MSCs at three different sites near the site of ligation. Animals with foot necrosis in the ischemia group were excluded from the analysis ($n = 2$) due to administration of a large bolus of pain relief and antibiotic agents that may affect the results of flow analysis. All animals were euthanized by intraperitoneal injection of 100 mg/kg pentobarbital.

Five groups of adult male ICR mice aged 12–16 weeks were used in this study: (1) intravenous administration of saline immediately after induction of ischemia and intramuscular administration of culture medium at day 7 (Ischemia group); (2) intravenous administration of saline immediately after induction of ischemia and intramuscular administration of $3 \times 10^6$ hiPSC-MSCs at day 7 (MSC-Saline group); (3) intravenous administration of $5 \times 10^5$ hiPSC-MSCs immediately after induction of ischemia and intramuscular administration of $3 \times 10^6$ hiPSC-MSCs at day 7 (MSC-MSC/once group); (4) repeated intravenous administration of $5 \times 10^5$ hiPSC-MSCs immediately and every week following induction of ischemia for 4 weeks and intramuscular administration of $3 \times 10^6$ hiPSC-MSCs at day 7 (MSC-MSC/week group); (5) repeated intravenous administration of $5 \times 10^5$ hiPSC-MSCs immediately and every 3 days following induction of ischemia for 4 weeks and intramuscular administration of $3 \times 10^6$ hiPSC-MSCs at day 7 (MSC-MSC/3 days group). Three mice in each group were sacrificed on day 7 and 14 for flow cytometry analysis of splenic immune cells and immunohistochemical staining to detect muscular macrophage infiltration. The remaining mice from five groups (ischemia group ($n = 8$), MSC-Saline group ($n = 10$), MSC-MSC/once group ($n = 10$), MSC-MSC/week group ($n = 8$), MSC-MSC/3 days group ($n = 8$)) were sacrificed on day 35. The fluorescent signal of the left ischemic limb was detected by an optical imaging system to evaluate cellular engraftment of the labeled hiPSC-MSCs. Blood perfusion of the hind limb was evaluated every week using a laser doppler imaging system[2,7].

**Laser Doppler imaging assessment of blood perfusion**. Blood perfusion of the hind limbs was assessed by serial Laser Doppler imaging analysis (Moor Instruments, Devon, UK) immediately and every week after induction of ischemia as described in our previous study[2]. In brief, all mice were first anesthetized by intraperitoneal injection of 100 mg/kg ketamine mixed with 10 mg/kg xylazine and then scanned under the laser head. All the animals laid on the stomach. Next, the digital color-coded images were analyzed to quantify blood flow over the ligated and non-ligated hind limbs from the knee to the toe. Blood perfusion was represented as the ratio of the values of the ligated limb versus the non-ligated limb and the mean values of three repeated images was calculated.

**Tissue collection and histological assessment**. The harvested tissues and organs were immediately fixed in ice-cold 4% buffered paraformaldehyde solution (AAJ19943K2, Thermo Fisher Scientific, NY, USA) and embedded in paraffin for sectioning into 5μm slices. Masson's trichrome staining was performed with a commercially available kit (AACSC009, American MasterTech Scientific, CA, USA) to identify fibrotic tissues. In brief, rehydrated tissue slices were first incubated with Bouin's fluid at 60 °C in a humidified atmosphere for 1 h. They were then stained with Weigert's mixed hematoxylin solution, Biebrich Scarlet/Acid Fuchsin solution and Aniline blue solution in sequence. Finally, they were fixed with 1% acetic acid solution. The degree of fibrosis was calculated by the ratio of fibrotic tissue (blue) to all muscular tissue. All the images captured and analyzed by AxioVision Rel. 4.5 software (Zeiss, GmbH, Oberkochen, Germany).

Immunohistochemical staining was performed to assess cell engraftment, neovascularization and infiltration of inflammatory cells. Double staining with anti-human mitochondria antibody (1:100, MAB1273, Merck Millipore, CA, USA) and anti-human GAPDH antibody (1:100, ab128915, Abcam, MA, USA) was applied to evaluate engraftment of the transplanted hiPSC-MSCs. The number of positive cells was counted over the peri-infarct regions in each treatment group and expressed as count per mm[2].

To evaluate neovascularization, immunohistochemical staining was performed with anti-mouse α-SMA (1:200, M0851, Sigma-Aldrich, MO, USA) and anti-mouse vWF (1:200, ab7356, Merck Millipore, CA, USA), respectively. Vessel density was determined as absolute number of positive staining vessels over the ischemic muscular regions in each treatment group and expressed as count per mm[2].

To assess infiltration of inflammatory cells, immunostaining of macrophages was also performed to estimate the number of leukocytes in the ischemic area. Macrophage infiltration was assessed by antibody staining with anti-mouse CD68 (1:200, ab31630, Abcam, MA, USA) independent of their polarization. The polarization of macrophages was further characterized by immunostaining with anti-Arginase 1 (1:100, ab60176, Abcam, MA, USA) for M2 phenotypes. Infiltration of macrophages was determined as the absolute number of CD68 positive macrophages over the ischemic regions in each treatment group and expressed as count per mm[2]. Polarization of macrophages was represented by the ratio of both CD68 and Arginase 1 positive M2 macrophages to CD68 positive total macrophages.

All data were measured in six random 40X fields from three different cross sections in a blinded fashion. Images of all sections were captured and analyzed by AxioVision Rel. 4.5 software (Zeiss, GmbH, Oberkochen, Germany).

**Cytokine profiling analysis**. The cytokine profiles of the limb tissue were quantified using a Mouse Inflammatory Factor Array (QAM-CYT-1, RayBiotech, Norcross, GA) to measure the level of inflammatory and anti-inflammatory factors, including IL-1A, IL-2, IL-10, IL-17A, VEGF, and MCSF. All the cytokine analysis were followed the manufacturer's instructions.

**Flow cytometry analysis**. Splenic Tregs and NK cells were analyzed by flow cytometry (Beckman-coulter, cytoflex-S[TM], FL, USA). After animals were sacrificed, the fresh spleen was harvested and washed with phosphate buffered saline (PBS) (SH30256.02, Cytiva, MA, USA). A single cell suspension of splenic lymphocytes was prepared by meshing through 100 μm nylon cell strainers (Falcon, NY, USA) and red blood cells removed by ACK lysing solution (A1049201, Invitrogen, CA, USA). One million splenocytes were then blocked with purified anti-mouse CD16/32 antibody (1:100, 101302, BioLegend, CA, USA) and 7AAD (1:50, 559925, BD Bioscience, NJ, USA) in flow buffer containing PBS, 1% BSA, 2 mM EDTA, 0.01% NaN3 for 10 min at 4 °C. After washing, these cells were subsequently used for staining of Tregs and NK cells. For Treg staining, a commercial Treg cells detection kit was used (130120674, Miltenyi Biotec, Cologne, Germany). In brief, one million splenocytes were stained with 1:20 anti-mouse CD4-VioBlue for 15 min and then fixed and made permeable with fixation-permeabilization buffers for 30 min on ice. Intracellular staining of Foxp3 was done by incubating with 1:20 anti-mouse Foxp3-PE antibody. NK cells were stained using 1:100 anti-mouse NK1.1-APC antibody (108709, Biolegend, CA, USA) and anti-mouse CD49b-FITC antibody (103503, Biolegend, CA, USA) for 30 min on ice. Stained cells were analyzed using a Cytomic FC500 flow cytometer (Beckman-Coulter, cytoflex-S[TM], FL, USA). The population of splenic Tregs and NK cells detected by flow cytometry was calculated from the percentage of cells in the number of viable cells, as showed in Supplementary Fig. 8. The number of Tregs was expressed as $Foxp3^+$ cells in a proportion of pre-gated $CD4^+$ cells. Flow images were analyzed by FlowJo software (V10.7.1, Becton, USA).

**The therapeutic efficacy of intravenous administration of hiPSC-MSCs**. First, to determine cell engraftment following a single cellular infusion, three mice were given a single intravenous administration of $5 \times 10^5$ hiPSC-MSCs at the same time as induction of hind-limb ischemia. Epi-fluorescent imaging was performed at 12 h and on day 3, 7 and 14.

To define the therapeutic efficacy of intravenous administration of hiPSC-MSCs, three more groups of mice were employed in this study: (1) intravenous administration of $5 \times 10^5$ hiPSC-MSCs immediately after induction of ischemia without intramuscular administration of hiPSC-MSCs (Saline-MSC/once group, $n = 8$); (2) repeated intravenous administration of $5 \times 10^5$ hiPSC-MSCs immediately and every week after induction of ischemia for 4 weeks without intramuscular administration of hiPSC-MSCs (Saline-MSC/week group, $n = 8$); (3) repeated intravenous administration of $5 \times 10^5$ hiPSC-MSCs immediately and

every 3 days after induction of ischemia for 4 weeks without intramuscular administration of hiPSC-MSCs (Saline-MSC/3 days group, $n = 8$). Serial Laser Doppler imaging analysis was performed immediately and every week after induction of ischemia to monitor the blood perfusion of hind limbs.

**The comparison of therapeutic efficacy of intravenous hiPSC-MSCs infusion and cyclosporine A injection**. To compare the therapeutic efficacy on cell survival and blood perfusion with hiPSC-MSCs and cyclosporine A (5 mg/kg, Sandimmune® Injection, Novartis, USA), three more groups of mice were employed in this study: (1) intravenous administration of saline immediately after induction of ischemia and intramuscular administration of $3 \times 10^6$ hiPSC-MSCs at day 7 combined with daily subcutaneous administration of cyclosporine A from day 7 to day 35 (MSC-Saline-Cyc group, $n = 3$); (2) intravenous administration of $5 \times 10^5$ hiPSC-MSCs immediately after induction of ischemia and intramuscular administration of $3 \times 10^6$ hiPSC-MSCs at day 7 combined with daily subcutaneous administration of cyclosporine A from day 7 to day 35 (MSC-MSC/once-Cyc group, $n = 3$); (3) repeated intravenous administration of $5 \times 10^5$ hiPSC-MSCs immediately and every week after induction of ischemia and intramuscular administration of $3 \times 10^6$ hiPSC-MSCs at day 7 with daily subcutaneous administration of cyclosporine A from day 7 to day 35 (MSC-MSC/week-Cyc group, $n = 3$). Fluorescent signal of the left ischemic limb was detected by an optical imaging system to evaluate the cellular engraftment of the labeled hiPSC-MSCs and blood perfusion of hind limbs was evaluated by Laser Doppler imaging system to assess the blood perfusion recovery every week.

**Statistics and reproducibility**. All data are expressed as mean ± SEM and analyzed SPSS software (SPSS, Inc., Chicago, IL, USA). Comparison between groups with serial changes at different time points was performed by two-way repeated ANOVA with Tukey post hoc test. The differences between groups within a single time point were compared by one-way ANOVA with Tukey post hoc test. All data were analyzed in a blinded manner and statistical significance was defined as a $p < 0.05$.

**Reporting summary**. Further information on research design is available in the Nature Research Reporting Summary linked to this article.

## Data availability
All the data are available from the corresponding authors (Dr. S.Y.L. or Prof. H.F.T.) upon reasonable request.

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

## Acknowledgements

This work was supported by the National Basic Research Program of China (973 Program, No. 2014CB965102), National Science Foundation of China (No: 31571407) and Hong Kong Research Grant Council GeneralResearch Fund (No. 17105319).

## Author contributions

S.J.S.: Experiment performance, data acquisition, data analysis and paper drafting; F.L.: Experiment performance, data acquisition, data analysis and paper drafting; M.D.: Experiment performance, data acquisition and data analysis; W-H.L.: Data acquisition and data analysis; W-Hon.L.: Experiment performance, data acquisition and data analysis; W-I.H.: Experiment performance and data acquisition; R.W.: Data acquisition and data analysis; Y.H.: Data analysis; S-Y.L: Study design, experiment performance, data analysis and paper revision; H-F.T.: Study design, data analysis and paper revision.

## Competing interests
The authors declare no competing interests.
