## [Peer Review File · Communications Biology]

Reviewers' comments:

Reviewer #1 (Remarks to the Author):

The authors of manuscript "Repeated Intravenous Administration of hiPSC-MSCs Enhance the Efficacy of Cell-based Therapy in Tissue regeneration" compare multiple regimens of MSC delivery in a model of hindlimb ischemia. Intramuscular delivery with repeated MSC IV delivery was found to be beneficial by assessment of blood perfusion, histological analysis of revascularization and fibrosis, and cell engraftment. The authors then evaluated the anti-inflammatory effects of these regimens to show similar benefits in groups with repeated IV delivery. Overall, the study was well conducted and presented clearly in figures and manuscript. Below are some recommendations to improve the manuscript.

Major considerations:

1. The structure of the result section can be much improved. I recommend the authors bring forward the supplemental data comparing IM injection vs IV injection of MSCs to justify examining the MSC delivery regimens in Figure 1. Also experiments for the purposes of optimizing can be moved to the methods section. I recommend that the results section ended with assessment of immunomodulation.
2. In the vWF staining, I would expect to see capillary formation where the lumen is not obvious, this usually occurs at the boundaries of myofibers. Did the authors assess capillary formation?
3. Figure 5 was not comprehensive in describing M1/M2 polarization. Using only one M2 marker is very weak evidence and difficult to make claims about favorable M2 polarization. At minimum, I would strongly recommend a comparison of an M1 marker and M2 marker to compare the ratio of M1/M2 macrophages. Authors can also consider staining for other inflammatory cytokines and anti-inflammatory cytokines to bolster their claim on M2 polarization.
4. There are samples at day 7, 14 and 35 for macrophage staining, but only day 35 for revascularization and fibrosis. I think there would be valuable information from earlier time points for revascularization and fibrosis, if the samples are available, it would be extremely helpful to the community. Even if the results are not significant, it should be included in the supplementary.
5. In the discussion, it would be of interest to readers to compare examples of bone marrow derived MSC from the literature to the iPSC derived MSCs in this study.

Minor considerations:

1. I find the transition from introduction to results to be jarring due to the numerous MSC treatment regimens. I would recommend the authors to take supplementary figure S1 as figure 1 to introduce these regimens. (I did not see in-text call out for the supplementary figure S1 in the manuscript)
2. For Figure S1, please label IV delivery and IM delivery explicitly to avoid confusion.
3. In the assessment of fibrosis, the authors wrote "Masson's Trichrome staining were performed to determine the percentage of fibrotic tissue in the ischemic limb". The authors should be more precise in describing how the percentages were measured? E.g. Percentage of trichrome staining 40 x image.

Reviewer #2 (Remarks to the Author):

The authors in this study, sought to demonstrate whether therapeutic efficacy can be improved by a combination of repeated intravenous administration and local transplantation of human induced pluripotential stem cell derived MSCs (hiPSC-MSCs).

To achieve this a mice model of hind-limb ischemia was established by ligation of left femoral artery. hiPSC-MSCs (5×10^5) was intravenously administrated immediately after induction of hind limb ischemia with or without following intravenous administration of hiPSC-MSCs every week or every 3 days. Intramuscular transplantation of hiPSC-MSCs (3×10^6) was performed one week after induction of hind-limb ischemia.

By comparing the therapeutic efficacy and cell survival of intramuscular transplantation of hiPSC-MSCs with or without a single or repeated intravenous administration of hiPSC-MSCs, they showed demonstrated increased splenic regulatory T cells (Tregs) activation, decreased splenic natural killer (NK) cells expression, increased.

They conclude that a combination of repeated systemic infusion and local transplantation of hiPSC-MSCs would improve the therapeutic efficacy of hiPSC-MSC- based therapy for cardiovascular disease.

In general the manuscript is very well written with clear cut supporting data.

I have minor comments

1. Although the authors have previously published the transplantation of hiPSC-MSCs in other disease scenarios it will be useful for the reader if they include data pertaining to the characterization of the derived hiPSC-MSCs
2. They should discuss potential paracrine effects of the transplanted cells
3. They stress immune-modulation as a potential mechanism- data showing hiPSC-MSC secreted cytokines should be shown in mouse serum
4. Can they exclude cell fusion?, this should at least be discussed.

Reviewer #3 (Remarks to the Author):

The authors report their findings that show improved therapeutic efficacy of weekly IM in combination with once, weekly or every 3-day IV hiPSC-MSC treatments in mouse model of ischemia reperfusion injury model. The outcome measures include tissue perfusion, neovascularization, decrease in fibrosis, number of splenic Tregs, NK cells and local polarization M2 macrophages at the site of ischemic and improved survival and engraftment of hiPSC cells.

The study was well designed and written but can be improved if the following can be clarified and addressed:

1. The experiment that compared repeated dosing of hiPSC-MSC which showed enhanced therapeutic efficacy of hiPSC-MSC is missing an important control group with IV hiPSC-MSC only (at a minimum one dose of 500k hiPSC-MSC). Without this control group the inference made Lane 131 -136 and Lane 169-172) are not completely accurate because it could be IM hiPSC-MSC enhancing IV hiPSC. From the results, it can only be concluded that combined IM/IV is better than single does of IM (MSC-saline group). This needs to be clarified. The reviewer understands a separate experiment was performed that showed IM hiPSC alone is better than IV hiPSC-MSC alone. This is helpful to know but should not lead to the assumption that the route of administration with weaker outcome is the one that was enhanced.
2. It is interesting that addition of Cyclosporin A did not make a difference. It was previously reported that immunosuppressive medications including cyclosporin affect MSC function, but the effect tend to vary depending the type of the drug and the dosage. -Durand et al J. Clin. Med. 2019, 8, 497; doi:10.3390/jcm8040497. A discussion of the under the Discussion section will have an added value especially when it comes to application, most patients are already receiving other medications.
3. Line 582 -I suspect intramyocardial injection is a typo, otherwise this does not make sense. Please correct/clarify.

Manuscript ID: COMMSBIO-21-2585

Manuscript TITLE: Repeated Intravenous Administration of hiPSC-MSCs Enhance the Efficacy of Cell-based Therapy in Tissue regeneration

Response to Reviewer 1:

The authors of manuscript “Repeated Intravenous Administration of hiPSC-MSCs Enhance the Efficacy of Cell-based Therapy in Tissue regeneration” compare multiple regimens of MSC delivery in a model of hindlimb ischemia. Intramuscular delivery with repeated MSC IV delivery was found to be beneficial by assessment of blood perfusion, histological analysis of revascularization and fibrosis, and cell engraftment. The authors then evaluated the anti-inflammatory effects of these regimens to show similar benefits in groups with repeated IV delivery.

Overall, the study was well conducted and presented clearly in figures and manuscript. Below are some recommendations to improve the manuscript.

Thanks for the comments from the Reviewer.

Major considerations:

1. The structure of the result section can be much improved. I recommend the authors bring forward the supplemental data comparing IM injection vs IV injection of MSCs to justify examining the MSC delivery regimens in Figure 1. Also experiments for the purposes of optimizing can be moved to the methods section. I recommend that the results section ended with assessment of immunomodulation.

As recommended, we have brought forward the data comparing IM injection vs. IV injection in Figure 1. The experiments for the purposes of optimizing have been moved to the methods section. Moreover, the results of immunomodulation have been brought backward and the results section ended with the comparison between hiPSC-MSC and cyclosporine A.

2. In the vWF staining, I would expect to see capillary formation where the lumen is not obvious, this usually occurs at the boundaries of myofibers. Did the authors assess capillary formation?

As recommended, capillary formation has been assessed by SMA and vWF staining at day 14 and day 35 in Figure 4 and the “Results section” (page 8, para 1, line 4-9):

“On day 14, intramuscular transplantation of hiPSC-MSCs in MSC-Saline group

did not increase arteriogenesis and capillary formation (Figure 4B-C, $p>0.05$). Nevertheless, systemic intravenous administration of hiPSC-MSCs in the MSC-MSC/once, MSC-MSC/week and MSC-MSC/3 days groups significantly improved arteriogenesis and capillary formation compared with the ischemia group (Figure 4B-C, all $p<0.05$).

3. Figure 5 was not comprehensive in describing M1/M2 polarization. Using only one M2 marker is very weak evidence and difficult to make claims about favorable M2 polarization. At minimum, I would strongly recommend a comparison of an M1 marker and M2 marker to compare the ratio of M1/M2 macrophages. Authors can also consider staining for other inflammatory cytokines and anti-inflammatory cytokines to bolster their claim on M2 polarization.

As recommended, the information on inflammatory cytokines and anti-inflammatory cytokines has been provided in Figure S3 and the “Methods Section” (page 25, para 1) and the “Results Section” (page 11, para 3 and page 12, para1-3):

“The cytokine profiles of the limb tissue were quantified using a Mouse Inflammatory Factor Array (QAM-CYT-1, RayBiotech, Norcross, GA) to measure the level of inflammatory and anti-inflammatory factors, including IL-1A, IL-2, IL-10, IL-17A, VEGF and MCSF. All the cytokine analysis were followed the manufacturer’s instructions.”

“The limb tissue level of a specific subset-related cytokines was measured using a commercial mouse inflammatory factor array. For anti-inflammatory cytokines, on day 14, there was no significant difference on interleukin (IL)-10 and vascular endothelial growth factor (VEGF) among the ischemia, MSC-Saline and MSC-MSC/once groups (Figure S3A-B, all $p>0.05$). Nonetheless, repeated systemic intravenous hiPSC-MSC infusion in the MSC-MSC/week and MSC-MSC/3 days groups significantly increased IL-10 and VEGF compared with the ischemia group (Figure S3A-B, all $p<0.05$). Moreover, an increase of IL-10 was observed in the MSC-MSC/week and MSC-MSC/3 days groups relative to the MSC-Saline group (Figure S3A-B, all $p<0.05$). On day 35, intramuscular transplantation of hiPSC-MSCs in the MSC-Saline group did not significantly improved IL-10 relative to ischemia group. Nevertheless, systemic intravenous hiPSC-MSC infusion in the MSC-MSC/once, MSC-MSC/week and MSC-MSC/3 days groups significantly improved IL-10 compared with the ischemia group (Figure S3A, all $p<0.05$). Moreover, repeated systemic intravenous hiPSC-MSC infusion in the MSC-MSC/week and MSC-MSC/3 days groups further increased IL-10 compared with the MSC-MSC/once group (Figure S3A, all $p<0.05$). No significant difference on VEGF was observed

among all five groups on day 35 (Figure S3B, all $p < 0.05$)."

"For inflammatory cytokines, on day 14, intramuscular transplantation of hiPSC-MSCs in the MSC-Saline, MSC-MSC/once, MSC-MSC/week and MSC-MSC/3 days groups significantly decreased IL-1A and IL-17A compared with the ischemia group (Figure S3C-D, all $p < 0.05$). Nonetheless, there was no significant difference among the MSC-Saline, MSC-MSC/once, MSC-MSC/week and MSC-MSC/3 days groups (Figure S3C-D, all $p > 0.05$). There was no significant difference on IL-2 and macrophage colony-stimulating factor (MCSF) among the ischemia, MSC-Saline and MSC-MSC/once groups (Figure S3E-F, all $p > 0.05$). Nonetheless, repeated systemic intravenous hiPSC-MSC infusion in the MSC-MSC/week and MSC-MSC/3 days groups significantly decreased IL-2 and MCSF compared with the ischemia group (Figure S3E-F, all $p < 0.05$). On day 35, intramuscular transplantation of hiPSC-MSCs in the MSC-Saline, MSC-MSC/once, MSC-MSC/week and MSC-MSC/3 days groups significantly reduced IL-17A relative to ischemia group (Figure S3D, all $p < 0.05$). Moreover, repeated systemic intravenous hiPSC-MSC infusion in the MSC-MSC/week and MSC-MSC/3 days groups further decreased IL-17A compared with the MSC-Saline and MSC-MSC/once groups respectively (Figure S3D, all $p < 0.05$). No significant difference on IL-1A, IL-2 and MCSF was observed among all five groups on day 35 (Figure S3C, S3E-F, all $p > 0.05$)."

"Taken together, our results demonstrated that systemic intravenous administration of hiPSC-MSCs could improve anti-inflammatory cytokines and decreased inflammatory cytokines. Repeated intravenous administration of hiPSC-MSCs every week or every 3 days further improved anti-inflammatory cytokines and decreased inflammatory cytokines compared with a single intravenous injection. No significant difference was observed between repeated intravenous administration of hiPSC-MSCs every week and every 3 days."

4. There are samples at day 7, 14 and 35 for macrophage staining, but only day 35 for revascularization and fibrosis. I think there would be valuable information from earlier time points for revascularization and fibrosis, if the samples are available, it would be extremely helpful to the community. Even if the results are not significant, it should be included in the supplementary.

As suggested, neovascularization and fibrosis on day 14 are now provided in Figure 4 and the "Results Section" (page 8, para 1, line 4-9 and page 8, para 2, line 3-7):

"On day 14, intramuscular transplantation of hiPSC-MSCs in MSC-Saline group did not increase arteriogenesis and capillary formation (Figure 4B-C, $p > 0.05$).

Nevertheless, systemic intravenous administration of hiPSC-MSCs in the MSC-MSC/once, MSC-MSC/week and MSC-MSC/3 days groups significantly improved arteriogenesis and capillary formation compared with the ischemia group (Figure 4B-C, all $p < 0.05$)."

"On day 14, intramuscular transplantation of hiPSC-MSCs in MSC-Saline group did not decrease fibrosis (Figure 4D, $p > 0.05$). Nevertheless, systemic intravenous administration of hiPSC-MSCs in the MSC-MSC/once, MSC-MSC/week and MSC-MSC/3 days groups significantly reduced fibrosis compared with the ischemia group (Figure 4D, all $p < 0.05$)."

5. In the discussion, it would be of interest to readers to compare examples of bone marrow derived MSC from the literature to the iPSC derived MSCs in this study.

As suggested, more information comparing BM-MSCs and hiPSC-MSCs are now provided in the "Discussion Section" (page 17, para 2, line 3-14):

"Pluripotent stem cell-derived cells with their unlimited cell source, such as hiPSC-MSCs, are a promising candidate for treatment of cardiovascular disease (7, 8). hiPSC-MSCs are not only morphologically similar to BM-MSCs, but also expressing similar antigens including CD44, CD49a, CD49e, CD73, CD105 and CD166 (7, 8). In addition, hiPSC-MSCs possess multiple immunomodulatory actions similar to BM-MSCs, suppressing proliferation, cytokine secretion and cytotoxicity of immune T-cells; and modulating the functions of Tregs and NK cells (7, 12). Our previous study showed that hiPSC-MSCs offered an unlimited "off-the-shelf" cell source with predictable therapeutic efficacy. In addition, we demonstrated that hiPSC-MSCs achieved a superior beneficial effect compared with BM-MSCs on induction of neovascularization due to their high engraftment rate and insensitivity to interferon (IFN)- γ -induced human leukocyte antigen (HLA) expression (7, 17)."

Minor points:

1. I find the transition from introduction to results to be jarring due to the numerous MSC treatment regimens. I would recommend the authors to take supplementary figure S1 as figure 1 to introduce these regimens. (I did not see in-text call out for the supplementary figure S1 in the manuscript)

As recommended, the Figure S1 is now moved to Figure 2.

2. For Figure S1, please label IV delivery and IM delivery explicitly to avoid confusion.

As suggested, we have labeled IV delivery and IM delivery in Figure 2.

3. In the assessment of fibrosis, the authors wrote “Masson’s Trichrome staining were performed to determine the percentage of fibrotic tissue in the ischemic limb”. The authors should be more precise in describing how the percentages were measured? E.g. Percentage of trichrome staining 40 x image.

As recommended, the detailed methods of histological assessment have been provided in the “Methods Section” (page 24, para 2 and page 25, para 2).

“The degree of fibrosis was calculated by the ratio of fibrotic tissue (blue) to all muscular tissue ”

“All data were measured in six random 40X fields from three different cross sections in a blinded fashion. Images of all sections were captured and analyzed by AxioVision Rel. 4.5 software (Zeiss, GmbH, Oberkochen, Germany).”

Response to Reviewer 2:

The authors in this study, sought to demonstrate whether therapeutic efficacy can be improved by a combination of repeated intravenous administration and local transplantation of human induced pluripotential stem cell derived MSCs (hiPSC-MSCs).

To achieve this a mice model of hind-limb ischemia was established by ligation of left femoral artery. hiPSC-MSCs (5x10⁵) was intravenously administered immediately after induction of hind limb ischemia with or without following intravenous administration of hiPSC-MSCs every week or every 3 days. Intramuscular transplantation of hiPSC-MSCs (3x10⁶) was performed one week after induction of hind-limb ischemia.

By comparing the therapeutic efficacy and cell survival of intramuscular transplantation of hiPSC-MSCs with or without a single or repeated intravenous administration of hiPSC-MSCs, they showed demonstrated increased splenic regulatory T cells (Tregs) activation, decreased splenic natural killer (NK) cells expression, increased.

They conclude that a combination of repeated systemic infusion and local transplantation of hiPSC-MSCs would improve the therapeutic efficacy of hiPSC-MSC- based therapy for cardiovascular disease.

Thanks for the comments from the Reviewer.

In general, the manuscript is very well written with clear cut supporting data.

Thanks for the comments from the Reviewer.

Minor points:

1. Although the authors have previously published the transplantation of hiPSC-MSCs in other disease scenarios it will be useful for the reader if they include data pertaining to the characterization of the derived hiPSC-MSCs.

As suggested, the characterization of hiPSC-MSCs has been discussed in the "Discussion Section" (page 17, para 2, line 3-14):

"Pluripotent stem cell-derived cells with their unlimited cell source, such as hiPSC-MSCs, are a promising candidate for treatment of cardiovascular disease (7, 8). hiPSC-MSCs are not only morphologically similar to BM-MSCs, but also expressing similar antigens including CD44, CD49a, CD49e, CD73, CD105 and CD166 (7, 8). In addition, hiPSC-MSCs possess multiple immunomodulatory actions similar to BM-MSCs, suppressing proliferation,

cytokine secretion and cytotoxicity of immune T-cells; and modulating the functions of Tregs and NK cells (7, 12). Our previous study showed that hiPSC-MSCs offered an unlimited “off-the-shelf” cell source with predictable therapeutic efficacy. In addition, our recent studies demonstrated that hiPSC-MSCs achieved a superior beneficial effect compared with BM-MSCs on induction of neovascularization due to their high engraftment rate and insensitivity to interferon (IFN)- γ -induced human leukocyte antigen (HLA) expression (7, 17).”

2. They should discuss potential paracrine effects of the transplanted cells.

As recommended, we have discussed the paracrine effects of the transplanted cells (*Discussion*, page 18, para 2).

“Previous studies demonstrated that MSC was a group of cells releasing a large spectrum of immunomodulatory factors as well as inhibiting a wide repertoire of pro-inflammatory factors (18). Our result confirmed the immunomodulatory effect of hiPSC-MSCs on cytokine release. In addition, our study further demonstrated that local cell transplantation only modulated a few inflammatory cytokines, such as IL-1A and IL-17A. Nonetheless, intravenous cell delivery has superior immunomodulatory efficacy on inflammatory and anti-inflammatory cytokines. Moreover, the immunomodulatory efficacy could be improved by repeated intravenous dosages. Taken together, our results showed intravenous hiPSC-MSC infusion modulate a wide variety of inflammatory and anti-inflammatory, which could not be achieved by local cell transplantation.”

3. They stress immune-modulation as a potential mechanism- data showing hiPSC-MSC secreted cytokines should be shown in mouse serum.

The methods and results of inflammatory cytokines and anti-inflammatory cytokines have been provided in the “Methods Section” (page 25, para 1) and the “Results Section” (page 11, para 3 and page 12, para1-3):

“The cytokine profiles of the limb tissue were quantified using a Mouse Inflammatory Factor Array (QAM-CYT-1, RayBiotech, Norcross, GA) to measure the level of inflammatory and anti-inflammatory factors, including IL-1A, IL-2, IL-10, IL-17A, VEGF and MCSF. All the cytokine analysis were followed the manufacturer's instructions.”

“The limb tissue level of a specific subset-related cytokines was measured using a commercial mouse inflammatory factor array. For anti-inflammatory

cytokines, on day 14, there was no significant difference on interleukin (IL)-10 and vascular endothelial growth factor (VEGF) among the ischemia, MSC-Saline and MSC-MSC/once groups (Figure S3A-B, all $p>0.05$). Nonetheless, repeated systemic intravenous hiPSC-MSC infusion in the MSC-MSC/week and MSC-MSC/3 days groups significantly increased IL-10 and VEGF compared with the ischemia group (Figure S3A-B, all $p<0.05$). Moreover, an increase of IL-10 was observed in the MSC-MSC/week and MSC-MSC/3 days groups relative to the MSC-Saline group (Figure S3A-B, all $p<0.05$). On day 35, intramuscular transplantation of hiPSC-MSCs in the MSC-Saline group did not significantly improved IL-10 relative to ischemia group. Nevertheless, systemic intravenous hiPSC-MSC infusion in the MSC-MSC/once, MSC-MSC/week and MSC-MSC/3 days groups significantly improved IL-10 compared with the ischemia group (Figure S3A, all $p<0.05$). Moreover, repeated systemic intravenous hiPSC-MSC infusion in the MSC-MSC/week and MSC-MSC/3 days groups further increased IL-10 compared with the MSC-MSC/once group (Figure S3A, all $p<0.05$). No significant difference on VEGF was observed among all five groups on day 35 (Figure S3B, all $p<0.05$)."

"For inflammatory cytokines, on day 14, intramuscular transplantation of hiPSC-MSCs in the MSC-Saline, MSC-MSC/once, MSC-MSC/week and MSC-MSC/3 days groups significantly decreased IL-1A and IL-17A compared with the ischemia group (Figure S3C-D, all $p<0.05$). Nonetheless, there was no significant difference among the MSC-Saline, MSC-MSC/once, MSC-MSC/week and MSC-MSC/3 days groups (Figure S3C-D, all $p>0.05$). There was no significant difference on IL-2 and macrophage colony-stimulating factor (MCSF) among the ischemia, MSC-Saline and MSC-MSC/once groups (Figure S3E-F, all $p>0.05$). Nonetheless, repeated systemic intravenous hiPSC-MSC infusion in the MSC-MSC/week and MSC-MSC/3 days groups significantly decreased IL-2 and MCSF compared with the ischemia group (Figure S3E-F, all $p<0.05$). On day 35, intramuscular transplantation of hiPSC-MSCs in the MSC-Saline, MSC-MSC/once, MSC-MSC/week and MSC-MSC/3 days groups significantly reduced IL-17A relative to ischemia group (Figure S3D, all $p<0.05$). Moreover, repeated systemic intravenous hiPSC-MSC infusion in the MSC-MSC/week and MSC-MSC/3 days groups further decreased IL-17A compared with the MSC-Saline and MSC-MSC/once groups respectively (Figure S3D, all $p<0.05$). No significant difference on IL-1A, IL-2 and MCSF was observed among all five groups on day 35 (Figure S3C, S3E-F, all $p>0.05$)."

4. Can they exclude cell fusion? This should at least be discussed.

Thanks for the reviewer point out this important point. Indeed, our study cannot exclude cell fusion. We have added a discussion in the "Discussion Section" (page 20, para 1, line 4-8).

“Fifth, consistent with previous studies, our study showed ischemia mice were benefited from the paracrine effects of hiPSC-MSC-based therapy. Our study cannot rule out the trans-differentiation and cell fusion of transplanted cells. Whether there is any trans-differentiation and cell fusion of transplanted cells after intramuscular transplantation need to be confirmed in the future experiment.”

Response to Reviewer 3:

The authors report their findings that show improved therapeutic efficacy of weekly IM in combination with once, weekly or every 3-day IV hiPSC-MSK treatments in mouse model of ischemia reperfusion injury model. The outcome measures include tissue perfusion, neovascularization, decrease in fibrosis, number of splenic Tregs, NK cells and local polarization M2 macrophages at the site of ischemic and improved survival and engraftment of hiPSC cells.

Thanks for the comments from the Reviewer.

The study was well designed and written but can be improved if the following can be clarified and addressed:

1. The experiment that compared repeated dosing of hiPSC-MSK which showed enhanced therapeutic efficacy of hiPSC-MSK is missing an important control group with IV hiPSC-MSK only (at a minimum one dose of 500k hiPSC-MSK). Without this control group the inference made Lane 131 -136 and Lane 169-172) are not completely accurate because it could be IM hiPSC-MSK enhancing IV hiPSC. From the results, it can only be concluded that combined IM/IV is better than single does of IM (MSK-saline group). This needs to be clarified. The reviewer understands a separate experiment was performed that showed IM hiPSC alone is better than IV hiPSC-MSK alone. This is helpful to know but should not lead to the assumption that the route of administration with weaker outcome is the one that was enhanced.

Thanks for the reviewer's comments. We have made changes below:

The result of intravenous hiPSC-MSK delivery groups was showed in the Figure 1 and "Results Section" (page 5, para 3):

"Three groups of mice that received intravenous hiPSC-MSK infusion once, every week or every 3 days without intramuscular administration of hiPSC-MSKs respectively were employed to assess the therapeutic efficacy of intravenous administration of hiPSC-MSKs (Figure 1A). Intravenous administration of hiPSC-MSKs once, every week or every 3 days without intramuscular administration of hiPSC-MSKs in the Saline-MSK/once, Saline-MSK/week and Saline-MSK/3 days groups significantly improved blood perfusion from day 7 onwards compared with the ischemia group (Figure 1B, all $p < 0.05$). Repeated intravenous administration of hiPSC-MSKs in the Saline-MSK/week and Saline-MSK/3 days groups further increased blood perfusion at day 35 compared with the Saline-MSK/once group (Figure 1B, all $p < 0.05$), although there was no difference between the first two groups (Figure 1B, $p > 0.05$). Nevertheless intramuscular administration of hiPSC-MSKs in the MSK-Saline group achieved a better beneficial effect than intravenous administration of hiPSC-MSKs in the Saline-MSK/once, Saline-MSK/week and

Saline-MSC/3 days groups from day 21 onwards (Figure 1B, all p<0.05). ”

We have revised Lane 131 -136 and Lane 169-172 (Results Section, page 7, para 2 and page 9, para 2):

“Taken together, our results showed that systemic intravenous administration of hiPSC-MSCs combined with intramuscular transplantation of hiPSC-MSCs improved blood perfusion in a mouse model of hind-limb ischemia relative to intramuscular hiPSC-MSC transplantation without systemic hiPSC-MSC delivery. In addition, repeated intravenous administration of hiPSC-MSCs every week or every 3 days further improved the therapeutic effects of hiPSC-MSC-based therapy compared with a single intravenous injection. No significant difference was observed between repeated intravenous administration of hiPSC-MSCs every week and every 3 days.”

“Taken together, our results showed that systemic intravenous administration of hiPSC-MSCs combined with intramuscular transplantation of hiPSC-MSCs promoted neovascularization and reduced fibrosis in a mouse model of hind-limb ischemia. Repeated intravenous administration of hiPSC-MSCs every week or every 3 days further increased the neovascularization and decreased the fibrosis following cellular transplantation compared with a single intravenous injection. No significant difference was observed between repeated intravenous administration of hiPSC-MSCs every week and every 3 days.”

We also have discussed the improved therapeutic efficacy of hiPSC-MSC-based therapy in “Discussion Section” (page 16, para 2, line 9-14)

“Second, a combination of single or repeated intravenous administration and local transplantation of hiPSC-MSCs significantly enhanced the therapeutic efficacy of MSC-based therapy. These improvements were associated with the immunomodulatory effect of intravenous cellular infusion as well as the increased cellular survival and engraftment of intramuscular cellular transplantation.”

2. It is interesting that addition of Cyclosporin A did not make a difference. It was previously reported that immunosuppressive medications including cyclosporin affect MSC function, but the effect tend to vary depending the type of the drug and the dosage. -Durand et al J. Clin. Med. 2019, 8, 497; doi:10.3390/jcm8040497. A discussion of the under the Discussion section will have an added value especially when it comes to application, most patients are already receiving other medications.

As recommended, a discussion of immunosuppressive drugs was added in the “Discussion Section” (page 19, para 2, line 23-28):

“Fourth, our study focused on investigating the immunomodulatory effect of

systemic intravenous hiPSC-MSC infusion. The underlying mechanism why subcutaneous administration of cyclosporine A did not improve cell engraftment in single or repeated intravenous hiPSC-MSC infusion groups remains unknown. Previous study demonstrated that the immunosuppressive effect of immunosuppressive medicines is vary depending on the types and the dosage of the drugs (27). In this study, we administered cyclosporine A at a dosage of 5mg/kg. Whether other immunosuppressive drugs or other dosage of cyclosporine A could improve cell engraftment under intravenous hiPSC-MSC infusion remains to be clarified in the future study.”

3. Line 582 -I suspect intramyocardial injection is a typo, otherwise this does not make sense. Please correct/clarify.

Thanks for the reviewer, the typo has been corrected.

“One week after induction of ischemia, animals were randomized to receive direct intramuscular injection of culture medium (100µl), or 3x10⁶ DiR labeled hiPSC-MSCs at three different sites near the site of ligation.” (page 22, para 2, line 11-14).

REVIEWERS' COMMENTS:

Reviewer #1 (Remarks to the Author):

The authors have address all concerns and recommendations. I commend the authors on their efforts and recommend this manuscript to be published.

Reviewer #2 (Remarks to the Author):

The authors have adequately responded to my questions.

Manuscript ID: COMMSBIO-21-2585A

Manuscript TITLE: Repeated Intravenous Administration of hiPSC-MSCs Enhance the Efficacy of Cell-based Therapy in Tissue Regeneration

Response to Reviewer 1:

The authors have addressed all concerns and recommendations. I commend the authors on their efforts and recommend this manuscript to be published.

Thanks for the comments from the Reviewer.

Response to Reviewer 2:

The authors have adequately responded to my questions.

Thanks for the comments from the Reviewer.

Response to Editor:

1) Please ensure that you upload the attached checklist when submitting your revision as well as your completed editorial policy checklist and the reporting summary document.

We have revised as recommended.

2) Please ensure that all authors in the author list have an affiliation and are included in the contribution section.

We have checked and revised.

3) Please ensure that all figures meet the minimum resolution requirements.

We have revised as recommended.

4) Please ensure that all figure panel labels are in lower case and that all corresponding text is updated accordingly.

We have checked and revised as recommended.

5) Please ensure that all figures have an accompanying legend (e.g. figure 2).

We have checked and revised.

6) Please add FACS gating strategy to the supplementary information PDF.

FACS gating strategy has been added in *Supplementary Figure 8*.

7) Please rename to your supplementary file to 'supplementary information'.

We have checked and revised as recommended.

8) Please ensure that all heat map figures (e.g. fig 1a) include a key.

We have revised as recommended.

9) Please see the attached table for guidance regarding source data underlying graphs – these should be provided in excel format rather than PDF.

We have revised as recommended.